# Reproducing FACTER: Fairness via Conformal Thresholding and Prompt Repair

**Oscar Miró López-Feliu**\*  *oscar.miro.lopez.feliu@student.uva.nl*
*University of Amsterdam*

**Daimy van Loo**\*  *daimy.van.loo@student.uva.nl*
*University of Amsterdam*

**Xanthos Kekkos**\*  *xanthos.kekkos@student.uva.nl*
*University of Amsterdam*

**Mikel Blom**\*  *mikel.blom@student.uva.nl*
*University of Amsterdam*

**Clara Rus**  *c.a.rus@uva.nl*
*University of Amsterdam*

**Reviewed on OpenReview:** *https://openreview.net/forum?id=4BPFVex4EM*

## Abstract

Fayyazi et al. (2025) recently proposed FACTER, a model-agnostic framework designed to jointly enforce fairness and statistical coverage in LLM-based recommendation through conformal thresholding and iterative prompt repair. In this work, we conduct a reproducibility study of the FACTER framework across diverse architectures and dataset sparsity levels, evaluating both the original open-ended generation task and a constrained re-ranking extension. Under the strict reproduction, we observe a divergence in recommendation utility, which we trace to underspecified target-set evaluation in the original study. We then use the constrained re-ranking setting to evaluate FACTER when the candidate set is fixed, and introduce a static Fair Zero-Shot baseline to isolate the contribution of the iterative prompt repair loop. Our analysis shows that FACTER consistently reduces adaptive-threshold violation counts, but that these reductions are not consistently reflected under the fixed threshold or in global fairness metrics. In the constrained ranking setting, static fairness instructions achieve comparable semantic-parity outcomes to FACTER's dynamic repair loop, suggesting that the additional online repair mechanism provides limited benefit in this formulation. All code and reproduction artifacts are available at `https://github.com/oscar-omlf/facter-repr`.

## 1 Introduction

Large Language Models (LLMs) are increasingly used as recommendation systems due to their ability to generate personalised item suggestions in natural language (Wu et al., 2024). Recent work explores LLM-based recommendations in domains such as movies and e-commerce, including unified text-to-text recommendation (e.g., P5; Geng et al., 2022), generative next-item frameworks (e.g., GPT4Rec; Li et al., 2023), and hybrid approaches that augment classical recommenders with LLM-generated signals (e.g., LLMRec; Wu et al., 2024). However, alongside their impressive capabilities, LLMs can inadvertently learn and amplify societal biases present in their training data (Bender et al., 2021). In a recommender system context, this implies that similar users from different demographic groups may receive systematically different recommendations,

---
\*Equal contribution

raising concerns of fairness and equity (Wang et al., 2023). Such unfairness can negatively affect multiple stakeholders, leading to reduced user satisfaction, skewed exposure for item providers, and increased regulatory risk for platforms.

A common approach to mitigating such biases relies on retraining or fine-tuning models under fairness constraints (Wang et al., 2023). In practice, retraining is not always feasible: many LLMs are deployed as black-box services where altering internal parameters is impractical. This motivates model-agnostic techniques to reduce bias using prompting and output-side control (Schick et al., 2021; Ma et al., 2023; Kamruzzaman & Kim, 2025). In this work, we conduct a reproducibility study of FACTER (Fayyazi et al., 2025), which proposes a black-box fairness framework for LLM-based recommendation. Building on an individual-fairness perspective (Dwork et al., 2012), FACTER expresses fairness through counterfactual stability, quantifying semantic variance in an embedding space, and calibrating a nonconformity threshold via conformal prediction (Angelopoulos & Bates, 2023). It then iteratively repairs the system prompt by extracting explicit "AVOID" constraints from detected violations.

Reproducing FACTER is practically important because it reports strong fairness improvements without the degradation in recommendation utility, which is otherwise a common trade-off (Wang et al., 2023). Moreover, its post-hoc, model-agnostic design is directly relevant to real-world deployments where only API access is available. Because several details required for an exactly matched implementation are absent or inconsistent across the paper and public code (detailed in Appendix A.1), this study should be read as a best-effort reproduction audit rather than a definitive adjudication of the original results. Given these ambiguities, we separate the evaluation into two parts: a best-effort reproduction of the original open-generation setting, and a controlled candidate re-ranking formulation that tests conformal thresholding and prompt repair under fixed candidate sets. The former documents the limits of reproducing the original utility claims, while the latter evaluates FACTER's fairness mechanisms in a setting where the candidate space is explicitly controlled.

Specifically, our study focuses on: (i) documenting the limits of reproducing the open-generation pipeline, (ii) evaluating the framework's behaviour under a distinct re-ranking extension, (iii) introducing a Fair Zero-Shot baseline to isolate the contribution of the iterative repair loop, and (iv) conducting a granular component analysis to assess how the observed violation reductions depend on adaptive threshold relaxation and prompt repair.

## 2 Scope of Reproducibility

### 2.1 Problem Setting

We study fairness interventions for LLM-based recommender systems under a *black-box* constraint: the system has access only to model inputs/outputs, not model parameters. In such settings, fairness mechanisms that rely on retraining, constrained optimisation, or gradient access are infeasible, motivating post-hoc interventions such as prompting, output monitoring, and filtering (Wang et al., 2023). Following FACTER (Fayyazi et al., 2025), we model the recommender as

$$\hat{y} = \text{LLM}(x, \boldsymbol{a}, \text{Prompt}_{sys}), \tag{1}$$

where $x$ denotes non-protected user context (e.g., interaction history), $\boldsymbol{a} = (a_1, \ldots, a_k)$ denotes a vector of protected attribute realisations (e.g., M (gender), under 18 (age), teacher (occupation)), and $\text{Prompt}_{sys}$ denotes the LLM system instructions. FACTER operationalises fairness as *counterfactual stability* with respect to $\boldsymbol{a}$: holding $x$ fixed, the *semantic content* of recommendations remains invariant to changes in $\boldsymbol{a}$. This intuition is closely related to counterfactual fairness, where outcomes should remain stable under interventions on protected attributes (Kusner et al., 2017). We briefly formalise the core components required for reproducibility in Section 3.

### 2.2 Core Claims

We evaluate FACTER against its core empirical claims (**C1-C4**), defining reproduction as matching effect direction and achieving comparable magnitudes within reasonable sensitivity to model choice, data sampling, and evaluation.

**C1 (Fairness effectiveness)** FACTER reduces fairness violations relative to a zero-shot baseline via an online loop that (i) detects threshold exceedances and (ii) injects "AVOID" prompt constraints mined from prior violations (Fayyazi et al., 2025).

**C2 (Limited loss in recommendation utility)** FACTER's fairness gains do not cause a disproportionate loss in recommendation utility relative to the zero-shot baseline (and UP5 where available), measured via top-$k$ ranking metrics (Recall@10, NDCG@10) (Fayyazi et al., 2025; Järvelin & Kekäläinen, 2002).

**C3 (Robustness across sparsity levels)** The approach remains effective across datasets with different sparsity and metadata characteristics, specifically MovieLens-1M and Amazon Movies & TV (Fayyazi et al., 2025; Harper & Konstan, 2016; Ni et al., 2019).

**C4 (Model-agnostic)** The intervention transfers across LLM backbones without model-specific finetuning, since it operates via prompting and output monitoring (Fayyazi et al., 2025).

## 3 Theoretical Framework

**Minimal-Attribute-Change (MAC) Fairness.** FACTER adopts an individual fairness notion (Dwork et al., 2012), where similar individuals must receive similar recommendations regardless of their sensitive attributes. Similarity is quantified via embeddings. A recommendation operator $\Gamma_{\text{fair}}$ satisfies MAC fairness if the semantic distance amidst outputs for users differing only in protected attributes ($\boldsymbol{a} \neq \boldsymbol{a}'$) is $\delta$-bounded:

$$\sup_{x,x':\rho(x,x')\leq\epsilon \,\wedge\, \boldsymbol{a}\neq\boldsymbol{a}'} \|\Gamma_{\text{fair}}(x,\boldsymbol{a}) - \Gamma_{\text{fair}}(x',\boldsymbol{a}')\| \leq \delta. \tag{2}$$

While the original paper presents $\rho$ as a distance metric, the implementation uses cosine similarity. We adhere to the implementation logic where semantic contiguity in embedding space proxies for theoretical similarity (Reimers & Gurevych, 2019; Chen et al., 2024).

**Conformal Calibration & Coverage.** To calibrate an acceptance threshold at level $(1 - \alpha)$, FACTER relies on split conformal prediction (Shafer & Vovk, 2008). A nonconformity score $S(x, \boldsymbol{a}, y)$ (Eq. 5) measures the violation of MAC principles. A threshold $Q_\alpha$ is calibrated on a held-out set such that:

$$\Gamma_{\text{fair}}(x,\boldsymbol{a}) = \{y \in \mathcal{Y} : S(x,\boldsymbol{a},y) \leq Q_\alpha\}. \tag{3}$$

In the online phase, $Q_\alpha^{(t)}$ is updated via an exponential decay mechanism based on observed violations (Eq. 6), dynamically resizing the acceptance region to handle distributional shifts.

## 4 Methodology

### 4.1 The FACTER Algorithm

We implemented FACTER from scratch, following the original paper where possible and resolving underspecified implementation details through the authors' public codebase and subsequent correspondence. Appendix A.1 documents the resulting implementation choices and paper–code discrepancies, particularly for the threshold update logic, string-matching tolerances, and multi-target relevance set definitions.

The framework is split into two phases: offline calibration, which estimates the initial threshold $Q_\alpha^{(0)}$ using the nonconformity score defined in Eq. 5, and online monitoring, in which the adaptive update rule (Eq. 6) is applied.

**Offline Calibration.** The objective of this phase is to establish an initial acceptance threshold $Q_\alpha^{(0)}$. We first encode user contexts and ground-truth items into a latent space using a sentence encoder. For calibration instance $x$, we compute a context embedding for user $i$: $e_i^x = f(x_i)$, where $f$ denotes the sentence encoder. The similarity matrix $W \in \mathbb{R}^{n \times n}$ is constructed to identify "counterfactual neighbours", defined as pairs of users who share similar interaction histories but differ in protected attributes. Similarity is determined by

cosine similarity $\cos(\cdot, \cdot)$, $\tau_x$ is the optional context-distance threshold described in the original formulation, and $\boldsymbol{a}_i$ is the protected attribute tuple (gender, age-bin, occupation):

$$W_{ij} = \begin{cases} \cos(e_i^x, e_j^x), & \text{if } \boldsymbol{a}_i \neq \boldsymbol{a}_j \text{ and } ||e_i^x - e_j^x||_2 \leq \tau_x \\ 0, & \text{otherwise} \end{cases} \tag{4}$$

Using these neighbourhoods, we compute a nonconformity score $S_i$ for each calibration instance, which serves as the conformity statistic which is used to detect violations and calibrate the initial threshold $Q_\alpha^{(0)}$. This score is a composite of the predictive error (deviation from the ground truth item) and a fairness penalty (the maximum semantic distance to recommendations given to neighbours).

$$S_i = \underbrace{1 - \cos(\text{Emb}(\hat{y}_i), e_i^y)}_{\text{Predictive Error } d_i} + \lambda \underbrace{\max_{j:W_{ij}>\tau_\rho} ||\text{Emb}(\hat{y}_i) - \text{Emb}(\hat{y}_j)||_2}_{\text{Fairness Penalty } \Delta_i} \tag{5}$$

The initial threshold $Q_\alpha^{(0)}$ is then set as the $(1 - \alpha)$-quantile of these scores, meaning that $(1 - \alpha)$ of the calibration data is accepted as "fair".

**Online Monitoring.** In the online phase, the system processes incoming queries sequentially. If a new recommendation's score $S_{new}$ exceeds the current threshold $Q_\alpha^{(t)}$, a violation is recorded. FACTER employs a dual-update mechanism to handle violations:

1. **Prompt Repair:** Violations are stored in a rolling buffer. Each violation instance yields a tuple $(g, Z)$ consisting of the user's demographic group $g$ and a set of extracted item features $Z$. We define a bias pattern as a pair $(g, z)$ where $z \in Z$ is a feature (e.g. a genre) associated with a violating recommendation. If a pattern $(g, z)$ appears at least $n$ times in the buffer, an explicit "AVOID: g $\rightarrow$ z" instruction is injected into the system prompt. For example, repeated violations recommending romance films to female users would yield the pattern $(g = \text{F}, z = \text{Romance})$.

2. **Threshold Adaptation:** Concurrently, the conformal threshold is updated via exponential smoothing when a violation occurs, to adapt the acceptance region in response to observed violations. Let $Q_\alpha^{(t)}$ denote the current threshold and $S_t$ the conformity score of the query:

$$Q_\alpha^{(t+1)} = \gamma Q_\alpha^{(t)} + (1 - \gamma)S_t \tag{6}$$

   with smoothing parameter $\gamma \in (0, 1)$ increasing the acceptance threshold in proportion to the severity of the observed violation.

## 4.2 Practical Instantiation

### 4.2.1 Datasets and Protected Attributes

We evaluate the reproducibility of FACTER using two datasets: MovieLens-1M (Harper & Konstan, 2016) and Amazon Movies & TV (McAuley et al., 2015; He & McAuley, 2016). To ensure controlled fairness evaluation, we use the same preprocessing pipeline for both datasets. All dataset preprocessing, filtering, and splitting procedures follow the original paper and released implementation without modification.

**General Processing Pipeline.** Across both datasets, we treat user ratings as interaction proxies and use chronological rating histories as context $x$. To reduce variability from data stochasticity (rather than modelling), we apply the following standardised steps:

1. **Filtration:** To ensure a robust preference signal for the recommendation task, we remove low-rated samples (rating $\leq 3$).

2. **Stochastic Control:** All random sampling, including protected attribute assignment and data splitting, is performed using a fixed seed (0).

3. **Sequence Construction:** User histories are time-ordered. We retain users with at least 10 interactions, sampling 10 sequential interactions as history. As clarified with the authors, a multi-target evaluation setting is consistent with the original study; because the exact target-set construction was not specified, we heuristically define the relevant target pool as the user's immediate next 10 reviews.

4. **Stratified Splitting:** Observations are partitioned into a 70/30 train-test split, providing calibration instances for the offline phase and evaluation instances for the online phase. We stratify by demographic groups to maintain balanced group representations.

**MovieLens-1M.** The ML-1M dataset contains ~1M ratings from 6k users over 4k movies on a 5-star scale (limited to whole-star ratings), as well as self-reported gender, binned age, and occupation as protected attributes. Following Fayyazi et al. (2025), bias patterns for prompt repairs are derived from movie titles and genres. We randomly sample 2,500 instances from the preprocessed data (1,750 calibration; 750 evaluation).

**Amazon.** The Amazon Movies & TV dataset (over 3.4M reviews) lacks protected attributes; following prior work, we synthetically assign age-bin, gender and occupation labels from the ML-1M attribute distribution. These attributes are used for fairness calibration, but do not reflect real user demographics. Thus, fairness evaluation on Amazon measures algorithmic stability under synthetic attribute perturbations rather than real-world demographic disparities. We sample 3,750 instances (2,625 calibration; 1,125 evaluation) due to sparsity. Prompt features use only movie titles, as the Amazon data lacks a canonical "genre" field.

### 4.2.2 Experimental Setup & Computational Resources

We evaluate three methods across both the open-ended generation and re-ranking task formulations: (i) a Neutral Zero-Shot baseline (Neutral), (ii) our Fair Zero-Shot baseline (Fair), and (iii) the FACTER algorithm. All experiments are repeated across three random seeds, and we report the mean and standard deviation for each metric.

FACTER is evaluated over three iterations per dataset and seed, with each iteration comprising a full test-set pass and retaining previously injected prompts and detected violations. Following the original study, we report results from the third iteration, while static baselines are evaluated in a single pass. To assess the framework's model dependence, we vary the underlying LLM and record performance on MovieLens-1M.

All experiments were conducted on a HPC cluster, using a node with an NVIDIA A100 GPU, 18 Intel Xeon CPU cores, and 120GB of RAM. A summary of the energy consumption and environmental impact metrics is reported in Section 6.6, and a detailed breakdown can be found in Appendix C.

### 4.2.3 Task Extension: Re-Ranking

In the original formulation, Fayyazi et al. evaluate recommendations in an open-ended generation setting: the LLM generates item titles without a constrained candidate set, and performance is measured by matching the output to a set of relevant items. This setup requires the model to implicitly retrieve the correct catalogue item from open generation. To disentangle fairness-aware ranking behaviour from open-ended catalogue search, we introduce a constrained re-ranking task in which the model ranks a fixed candidate pool.

**Task Formalization.** We formulate recommendation as candidate re-ranking. Given a candidate set $\mathcal{C}$ containing a set of relevant ground-truth items $\mathcal{R}_{\text{rel}}$ (multi-target) and $|\mathcal{C}| - |\mathcal{R}_{\text{rel}}|$ uniformly sampled negatives (excluding user history), the model outputs a ranked list. We evaluate the top $k = 10$ items $\mathcal{R}$. Candidates are shuffled per query to prevent positional and "lost-in-the-middle" biases. This setup assumes successful candidate retrieval. In practice, a high-recall retriever (e.g., BM25 (Robertson et al., 1994) or vector search) reduces the corpus to a manageable subset. By including the relevant set $\mathcal{R}_{\text{rel}}$ in $\mathcal{C}$, we decouple evaluation from retriever-specific limitations and focus on the LLM's fairness-aware ranking behaviour.

**Algorithmic Adaptation.** We adapt FACTER to this constrained setting by adding ranking-specific instructions and the explicit candidate list to the prompt (Appendix A.4). This lets us test conformal thresholding and prompt repair when the search space is fixed, rather than open-ended generation.

### 4.2.4 Models and Implementation

To evaluate sensitivity to the underlying language model, we use three open-weight LLM backbones: Llama 2-7B (Touvron et al., 2023), Llama 3-8B (Grattafiori et al., 2024), and Mistral-7B (Jiang et al., 2023). We use the fine-tuned MPNet sentence transformer from the original study[1] for all semantic representations. Our Neutral baseline uses standard recommendation prompts without fairness instructions, while our Fair baseline uses static fairness-aware prompting without conformal adaptation. This allows us to compare FACTER against both an unconstrained zero-shot baseline and a prompt-only fairness baseline; full prompt templates are provided in Appendix A.4.

The original study used UP5 (Hua et al., 2024) as its primary fairness-aware baseline. UP5 relies on parameter-efficient prefix tuning, making it a distinct comparator rather than a direct ablation of FACTER's conformal thresholding and prompt-repair mechanisms. However, due to missing pre-trained checkpoints and code incompatibilities in the public UP5 repository, a functional reproduction was not feasible within this study; Appendix A.7 provides a technical breakdown. We therefore report the UP5 results from the original manuscript only as reference values, and base our direct comparisons and examination of the core claims on the Neutral, Fair, and FACTER methods evaluated under our implementation.

### 4.2.5 Hyperparameters

We do not perform additional hyperparameter search and adopt the reported tuned values from the original study: fairness penalty $\lambda = 0.7$, threshold decay $\gamma = 0.95$, similarity threshold $\tau_\rho = 0.9$, conformal level $\alpha = 0.1$, and FIFO violation buffer size $M = 50$ with rule-mining threshold $n = 3$. Embedding and generation batch sizes were selected based on hardware constraints (see Appendix A.3 full list of hyperparameters).

Consistent with the authors' released implementation, we do not enforce the context similarity threshold $\tau_x$, and constrain "fair" neighbourhoods through $\tau_\rho$. We map LLM outputs to catalogue items using a cosine similarity threshold of 0.65 and fix the recommendation list length to $k = 10$.

Parameters introduced by our ranking-based extension were determined heuristically and held constant across experiments. The candidate set size $|\mathcal{C}| = 40$ was chosen to provide sufficient ranking difficulty while remaining computationally feasible given our available compute budget.

## 4.3 Evaluation

Our evaluation uses a dual-objective framework to assess both fairness and recommendation utility. We align with the authors' released implementation where possible, noting that certain implementation choices differ from the mathematical formalisations in the paper; Appendix A.2 summarises these differences.

**Fairness Metrics.** For fairness evaluation, FACTER defines bias indicators as instances of large semantic changes in generated outputs when only protected attributes are perturbed. It therefore defines a minimal-attribute-change property: modifying only the sensitive attribute $\boldsymbol{a} \rightarrow \boldsymbol{a'}$ while holding non-protected features $x$ fixed should not yield large output discrepancies, measured via embedding-space distance.

The *Sensitive-to-Neutral Similarity Ratio (SNSR)* metric summarises group-level disparities, while the *Counterfactual Fairness Ratio (CFR)* assesses responsiveness to counterfactual flips of protected attributes. When combined, local fairness controls and global aggregate metrics provide a more complete picture of fairness behaviour (Fayyazi et al. (2025)).

Let $\mathcal{G}$ be the set of protected groups and $\mathbf{h}_g$ the centroid of pooled recommendation embeddings for group $g \in \mathcal{G}$. We define the pairwise cosine distance as $d_{\cos}(g_i, g_j) = 1 - \mathbf{h}_{g_i} \cdot \mathbf{h}_{g_j}$. SNSR is calculated over the set

---

[1]https://huggingface.co/JJTsao/fine-tuned_movie_retriever-all-mpnet-base-v2

of all unique group pairs $\mathcal{P}$ as:

$$\text{SNSR} = \max_{\{g_i, g_j\} \in \mathcal{P}} d_{\cos}(g_i, g_j) \tag{7}$$

Although lower values indicate more uniform treatment, the original implementation restricts evaluation to multi-attribute groups with $n \geq 30$. This threshold, combined with the pooling of per-example Top-$k$ embeddings prior to group averaging, can over-smooth results and mask fine-grained disparities. Furthermore, this restriction poses a structural risk: the metrics may become incomputable if the dataset lacks sufficient groups meeting this requirement. We address this by computing SNSR per attribute and on the multi-attribute intersection. Additionally we evaluate the CFR, defined as the average distance between original and counterfactually perturbed recommendation embeddings:

$$\text{CFR} = \mathbb{E}_{x \sim \mathcal{D}} \left[ \| f_\phi(\hat{y}) - f_\phi(\hat{y}_{\neg s}) \|_2 \right] \tag{8}$$

We use $L_2$ distance for CFR to match the paper's definition. By contrast, the SNSR metric follows the cosine-distance proxy used in the released implementation, because the paper's SNSR formulation is not directly reproducible from the released artifacts. Our CFR implementation supports both multi- and single-attribute assessments. The former involves flipping the gender attribute and resampling age and occupation to distinct values to form a valid counterfactual state, the latter modifies a single attribute ceteris paribus.

Furthermore, we monitor the *Violation Count* to assess the reliability of the thresholding mechanism. This metric represents the number of instances where the non-conformity score $S_i$ exceeds the threshold $Q_\alpha$. Given the ambiguity in the original study regarding the precise point of measurement, we report the violation count using two distinct thresholds: the sample-corrected post-calibration threshold $(Q_\alpha^{(0)} + \frac{C}{\sqrt{n}})$, and the dynamic threshold $(Q_\alpha^{(t)})$. This reporting ensures a comprehensive view of how the model manages the fairness-utility trade-off across different stages of the adaptive process.

**Utility and Calibration Metrics.** We report *Recall@k*, and *NDCG@k* (Järvelin & Kekäläinen, 2002),

$$\text{Recall@k} = \frac{|\mathcal{R}_{\text{rel}} \cap \mathcal{R}_k|}{|\mathcal{R}_{\text{rel}}|}, \quad \text{and} \quad \text{NDCG@k} = \frac{1}{\text{IDCG@k}} \sum_{r=1}^{k} \frac{2^{\text{rel}(r)} - 1}{\log_2(r+1)}. \tag{9}$$

where $\mathcal{R}_{\text{rel}}$ is the set of relevant target items. Additionally we compute *Valid@k*, which measures the fraction of the top-k generated recommendations that can be mapped to a valid catalogue item in our database.

# 5 Results

We assess the reproducibility of FACTER by comparing its original open-ended generation setting with our re-ranking extension, and by isolating adaptive thresholding and dynamic prompting. Accordingly, the open-generation results are used to assess the original claims, whereas the re-ranking results are used to analyze a separate extension. UP5 values are reported only as reference values from the original study, as reproduction was not possible. Across all tables, bold values indicate the best-performing method per dataset and metric within each column group, and reproduced values are reported as mean (SD) over three random seeds.

## 5.1 Original Setup: Reproducibility of the Open-Ended Generation Task

**Utility Divergence.** We observe a gap in recommendation utility under the strict open-ended generation protocol. As shown in Table 1, our reproduction yields near-zero NDCG@10 and Recall@10 values on both datasets, contrasting with the strong results reported in the original study. FACTER's marginally lower scores relative to the zero-shot baseline align with the original finding of limited utility loss, though the absolute magnitudes suggest a potential floor effect caused by the unconstrained search space. This limitation is further evidenced by the Valid@k metric, which indicates that 88–90% of generated recommendations on Amazon and 72–75% on ML-1M can be mapped back to valid catalogue items (see Appendix B.3), suggesting that invalid identifiers and hallucinated catalogue items may contribute to the observed utility gap.

Table 1: Reproduction (Open-Ended Generation): Original (**O**) vs. Reproduced (**R**) results using **Llama-3-8B**. *Neutral:* neutral zero-shot, *Fair:* fair zero-shot baseline. FACTER scores are reported at iteration 3.

| Data | Model | NDCG@10 ↑ | | Recall@10 ↑ | | SNSR[1] ↓ | | CFR ↓ | | Constraint Violations ↓ | | |
|------|-------|-----------|-----|-------------|-----|-----------|-----|-------|-----|------|------|------|
| | | **O** | **R** | **O** | **R** | **O** | **R** | **O** | **R** | **O** | **R** $(V_{Q_\alpha}^{(t)})$[4] | **R** $(V_{Q_\alpha}^{(0)})$ |
| Movie Lens 1M | Neutral | **.458** | **.035 (.001)** | **.402** | **.031 (.001)** | .083 | .037 (.018) | .742 | **.667 (.005)** | 112 | – | 42.7 (5.8) |
| | Fair[2] | – | .030 (.002) | – | .027 (.002) | – | .038 (.008) | – | .683 (.004) | – | – | 41.0 (4.4) |
| | UP5[3] | .427 | – | .381 | – | .049 | – | .613 | – | 28 | – | – |
| | **FACTER** | .445 | .029 (.001) | .389 | .026 (.000) | **.041** | **.032 (.005)** | **.591** | .702 (.009) | **5** | **9.0 (0.0)** | **35.3 (6.4)** |
| Amazon Movies & TV | Neutral | **.351** | **.004 (.000)** | **.317** | **.005 (.001)** | .121 | – | .814 | **.879 (.018)** | 198 | – | 107.7 (1.5) |
| | Fair[2] | – | **.004 (.001)** | – | .004 (.000) | – | – | – | .906 (.023) | – | – | **91.3 (8.4)** |
| | UP5[3] | .328 | – | .294 | – | .067 | – | .721 | – | 63 | – | – |
| | **FACTER** | .339 | .003 (.001) | .301 | .003 (.000) | **.053** | – | **.634** | .922 (.001) | **18** | **3.0 (1.7)** | 110.7 (11.0) |

[1]SNSR on Amazon could not be reported due to the multi-attribute group constraint (see appendix B for more details).
[2]Fair scores are not reported in the original results as the Fair zero-shot baseline was not implemented.
[3]UP5 scores are not reported in the reproduction due to missing checkpoints and code incompatibilities.
[4]Violations under $V_{Q_\alpha}^{(t)}$ are only reported for FACTER, as the static baselines do not employ an iterative threshold update.

**Fairness Trends.** Despite the low absolute utility in open generation, several fairness-related trends align qualitatively with the original study. In particular, FACTER reduces adaptive-threshold violations from 42.7 (Neutral) to 9.0 on ML-1M, and from 107.7 to 3.0 on Amazon, when measured via the adaptive threshold $(V_{Q_\alpha}^{(t)})$. FACTER also achieves the lowest SNSR on ML-1M (0.032), although the decrease relative to the Neutral (0.037) and Fair (0.038) baselines is considerably smaller than in the original results. At the same time, counterfactual stability does not improve in our reproduction: FACTER yields a higher CFR (0.702) than the Neutral baseline (0.667) on ML-1M, with an even more pronounced gap on Amazon. SNSR could not be reported for Amazon under the multi-attribute group constraint; Appendix B provides additional discussion and single-attribute results.

Table 2: Reproduction (Open-Ended Generation): Model-wise comparison on ML-1M. **O**: original, **R**: reproduced (**O** values for the Neutral baseline are not provided in original study)

| Architecture | Model | NDCG@10 ↑ | | Recall@10 ↑ | | SNSR ↓ | | CFR ↓ | | Constraint Violations ↓ | | |
|--------------|-------|-----------|-----|-------------|-----|--------|-----|-------|-----|------|------|------|
| | | **O** | **R** | **O** | **R** | **O** | **R** | **O** | **R** | **O** | **R** $(V_{Q_\alpha}^{(t)})$ | **R** $(V_{Q_\alpha}^{(0)})$ |
| LLaMA3-8B | Neutral | - | **.035 (.001)** | - | **.031 (.001)** | - | .037 (.018) | - | **.666 (.005)** | - | – | 42.7 (5.8) |
| | FACTER | .440 | .029 (.001) | .383 | .026 (.000) | .039 | **.032 (.005)** | .576 | .702 (.009) | 3 | **9.0 (0.0)** | **35.3 (6.4)** |
| LLaMA2-7B | Neutral | - | **.028 (.003)** | - | **.026 (.003)** | - | **.017 (.002)** | - | **.640 (.004)** | - | – | **45.3 (9.1)** |
| | FACTER | .444 | .022 (.002) | .391 | .020 (.002) | .041 | .023 (.008) | .595 | .712 (.016) | 5 | **8.3 (0.6)** | 60.0 (9.6) |
| Mistral-7B | Neutral | - | **.031 (.002)** | - | **.026 (.001)** | - | .052 (.007) | - | **.741 (.021)** | - | – | 39.3 (7.5) |
| | FACTER | .451 | .026 (.001) | .397 | .022 (.001) | .043 | **.041 (.005)** | .602 | .785 (.031) | 7 | **7.3 (1.2)** | **37.0 (4.4)** |

**Model Agnosticism.** The behavioural trends observed on ML-1M are qualitatively similar across the three tested backbones (Llama 3-8B, Llama 2-7B, and Mistral-7B), providing partial empirical support for FACTER's model-agnostic design (Table 2). Across all backbones, FACTER's utility is uniformly low (NDCG@10 confined to .022–.029) and consistently trails the Neutral baseline. This consistency extends to the fairness metrics: FACTER achieves a reduction in adaptive constraint violations $(V_{Q_\alpha}^{(t)})$ to between 7.3 and 9.0, compared to the baseline range of 39.3 to 45.3. While absolute utility diverges from the original study, the consistent differences between FACTER and the baselines suggest a similar performance profile across the tested architectures.

**Iterative Fairness Refinement.** We observe a continuous reduction in adaptive-threshold violations $(V_{Q_\alpha}^{(t)})$ over iterations, with violations decreasing monotonically across the three evaluation passes (Figure 1). While the original study reports a steep decline from initially high violation counts, our reproduction begins with a lower initial count, resulting in a shallower but consistent downward trajectory. However, this apparent iterative refinement is strictly confined to FACTER's internal conformal bounds. As detailed in Table 1, this

step-wise reduction in internal violations does not yield commensurate improvements in structural fairness metrics, as CFR and SNSR fail to meaningfully improve over the zero-shot baselines by the final iteration.

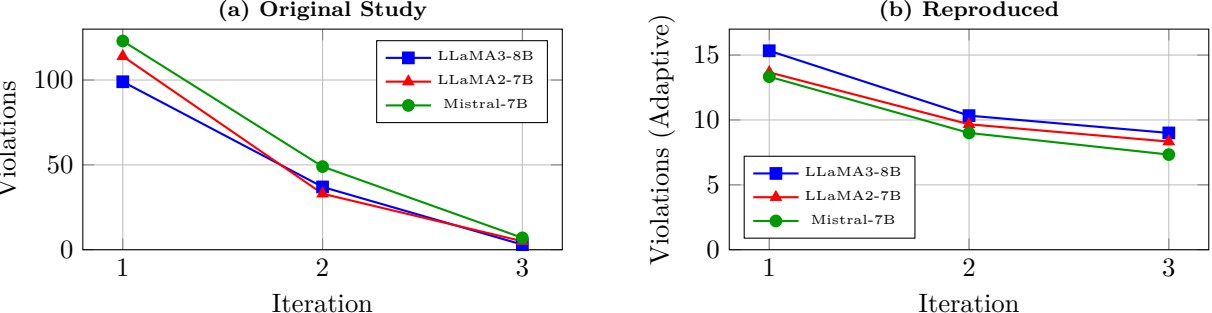

Figure 1: Violation reduction over iterations on ML-1M for the open-generation task across the three tested LLM backbones. **(a)** Original (taken directly from Fayyazi et al. (2025)) shows steep initial decline. **(b)** Reproduction shows a lower start with a shallower decrease.

## 5.2 Evaluation under Re-Ranking Extension

**Utility under Re-Ranking.** Evaluating the models in the controlled re-ranking setting yields recommendation utility metrics that are markedly higher than those observed in the open-ended generation setting. As detailed in Table 3, by constraining the search space to a fixed candidate list, the Neutral baseline achieves an NDCG@10 of 0.466 and a Recall@10 of 0.433 on the ML-1M dataset. FACTER closely follows this performance, yielding an NDCG@10 of 0.438 and a Recall@10 of 0.407. This suggests that the underlying LLMs can identify relevant items more reliably when the task is reformulated as a discriminative ranking problem.

**Fairness Preservation.** Within this higher-utility re-ranking regime, FACTER maintains low adaptive violation counts ($V_{Q_\alpha}^{(t)}$). On the ML-1M dataset, adaptive violations under FACTER drop to 7.7, compared to 55.7 fixed-threshold violations observed in the static Fair baseline. A similar trend is observed on the Amazon dataset, where FACTER yields only 1.7 adaptive violations. However, in contrast to the reduction in these internal violations, global proxy fairness metrics such as SNSR and CFR show marginal differences across the evaluated methods. For instance, FACTER yields a CFR of 0.553 on ML-1M, which is slightly worse than both the Neutral (0.533) and Fair (0.534) baselines. These results suggest that satisfying FACTER's adaptive conformal bounds does not necessarily translate into improved group-level semantic parity or counterfactual stability relative to static baselines.

Table 3: Extension (Re-Ranking): Performance and Fairness on ML-1M and Amazon datasets using the **Llama-3-8B** backbone. FACTER scores are reported at iteration 3.

| Dataset | Model | NDCG@10 ↑ | Recall@10 ↑ | SNSR [1] ↓ | CFR ↓ | $V_{Q_\alpha}^{(t)}$ [2] ↓ | $V_{Q_\alpha}^{(0)}$ ↓ |
|---|---|---|---|---|---|---|---|
| ML-1M | Neutral | **.466 (.002)** | **.433 (.004)** | .026 (.003) | **.533 (.007)** | – | 61.3 (7.1) |
| | Fair | .456 (.006) | .426 (.004) | **.024 (.006)** | .534 (.006) | – | **55.7 (4.9)** |
| | **FACTER** | .438 (.007) | .407 (.005) | **.024 (.003)** | .553 (.010) | **7.7 (1.5)** | 60.3 (4.2) |
| Amazon Movies & TV | Neutral | .448 (.005) | .419 (.007) | – | .715 (.028) | – | 92.0 (4.4) |
| | Fair | **.458 (.003)** | **.431 (.003)** | – | **.681 (.012)** | – | 91.3 (3.2) |
| | **FACTER** | .451 (.004) | .423 (.002) | – | .707 (.005) | **1.7 (1.2)** | **88.7 (5.8)** |

[1]SNSR on Amazon could not be reported due to the multi-attribute group constraint (see appendix B for more details).

[2]Violations under $V_{Q_\alpha}^{(t)}$ are only reported for FACTER, as the static baselines do not employ an iterative threshold update.

### 5.3 Deconstructing the Fairness Mechanism

**Threshold Dynamics.** The reported reduction in FACTER's violations emerges as highly threshold-dependent (Tables 1 and 3). In the open-generation task on ML-1M, FACTER yields 9.0 violations under the adaptive threshold $(V_{Q_\alpha}^{(t)})$, but this figure increases to 35.3 under the fixed, population-calibrated threshold $(V_{Q_\alpha}^{(0)})$, closely aligning with the Neutral (42.7) and Fair (41.0) baselines. This discrepancy is even more pronounced on the Amazon dataset, where FACTER's adaptive violations drop to 3.0, while its fixed-threshold violations reach 110.7 (compared to 107.7 for Neutral). This pattern of low adaptive but high fixed violations that mirror the baselines persists across both datasets in the re-ranking formulation, suggesting that the apparent reduction is largely tied to the moving threshold.

**Dynamic vs. Static Prompting** We compare the fully dynamic FACTER loop against our Fair Zero-Shot baseline to isolate the impact of the iterative repair mechanism in the re-ranking setting (Table 3). In the ML-1M re-ranking formulation, the static Fair baseline achieves an SNSR of 0.024 and 55.7 fixed-threshold violations $(V_{Q_\alpha}^{(0)})$, performing comparably to the fully dynamic FACTER loop (0.024 SNSR and 60.3 violations under $V_{Q_\alpha}^{(0)}$). Similarly, on the Amazon dataset, the Fair baseline yields a CFR of 0.681 and 91.3 fixed violations, while FACTER results in a CFR of 0.707 and 88.7 fixed violations. These results suggest broadly comparable SNSR, CFR, and fixed-threshold violation outcomes between the static baseline and the dynamic online monitoring loop.

## 6 Discussion

Our reproduction of FACTER yields a nuanced verdict. We find support for the use of conformal thresholding to dynamically bound internal violation counts, but observe substantial divergences in recommendation utility and mixed evidence for the practical necessity of the online prompt repair loop. This section deconstructs these findings, specifically addressing the utility gap, threshold dynamics, and the broader reproducibility landscape.

### 6.1 The Utility Gap: Evaluation Protocol vs. Task Difficulty

The gap between the reported open-generation utility and our strict reproduction highlights the inherent difficulty of zero-shot recommendation, under which the model must not only search a large item space but also emit titles that can be matched to catalogue identifiers (Wang & Lim, 2023; Geng et al., 2022; Li et al., 2023; 2024). However, task difficulty alone likely does not fully explain the magnitude of the divergence, given our matched model backbones. Our analysis suggests that the gap is partly tied to underspecified evaluation choices, including string-matching tolerances and the construction of the multi-target relevance set. If the original study employed a more permissive definition of relevance than our heuristic next 10 reviews, it would widen the target space and raise measured utility. Thus, while we cannot recover the open-generation utility from the public artefacts alone, this does not invalidate the original findings; rather, it shows that FACTER's empirical utility is sensitive to evaluation protocol. The higher utility observed in our re-ranking extension is consistent with this interpretation, but because the extension changes the task formulation, it does not by itself reconcile the original open-generation result. We therefore treat re-ranking as a distinct diagnostic setting for analyzing FACTER's fairness mechanisms under a controlled candidate space.

### 6.2 Threshold Dynamics: Risk Management vs. Bias Erasure

We initially hypothesised that the persistence of violations under the fixed threshold stemmed from a "null penalty" artefact, where strict neighbourhood definitions rendered the fairness penalty ($\Delta$) negligible. However, component analysis (Appendix B, Table 9) refutes this: the fairness penalty actively contributes approximately 50% to the non-conformity score of violating instances (e.g., 49.65% in Iteration 1).

The persistence of violations under the fixed threshold $(Q_\alpha^{(0)})$ is therefore not a mathematical artefact, but a structural limitation of the prompt repair mechanism. Specifically, dynamically injected "AVOID"

constraints fail to reduce conformity scores. Over three iterations, the average non-conformity score for violations increases from 1.492 to 1.703. Decomposing this score reveals that while predictive error ($d$) marginally improves ($0.881 \rightarrow 0.804$), the fairness penalty worsens significantly ($0.872 \rightarrow 1.283$). This indicates that the LLM prioritises predictive confidence over the injected fairness constraints.

Consequently, FACTER appears to satisfy its statistical coverage guarantee primarily by adapting the threshold $Q_\alpha^{(t)}$, which reduces the number of violations recorded under the dynamic threshold, rather than by clearly improving behavior under the fixed baseline $Q_\alpha^{(0)}$.

This pattern is consistent with the weaker performance observed under external proxy metrics such as CFR and SNSR. The explicit negative constraints injected during prompt repair may destabilize counterfactual stability, pushing the model toward semantically different catalogue regions when demographic inputs are perturbed. In our results, the dynamic repair loop achieves lower adaptive-threshold violation counts, but does not clearly improve group-level semantic parity or counterfactual stability relative to static fairness instructions.

## 6.3 Efficacy of Static vs. Dynamic Prompting

Our comparative ablation within the re-ranking formulation indicates that the iterative prompt repair loop provides limited fairness improvements over static instructions. The similarity between the static Fair baseline and the full FACTER loop suggests that in the constrained re-ranking setting, the explicitness of a clear static base instruction accounts for much of the observed effect.

Consequently, the significant computational overhead and inference latency introduced by FACTER's online monitoring module may yield diminishing returns in discriminative ranking contexts. We hypothesise that while the dynamic repair mechanism may be less useful for these constrained settings, it may still possess utility in unconstrained generative tasks where stochastic drift is higher (Borah et al., 2025), provided the underlying open-generation utility gap can be overcome. Finally, the observed utility degradation in the Fair Zero-Shot baseline relative to the Neutral baseline is consistent with an "alignment tax" (Korkmaz et al., 2025), suggesting that strict fairness constraints can compete with relevance signals regardless of whether they are applied statically or dynamically.

## 6.4 Reproducibility Constraints

FACTER's training-free design makes it easier to reproduce than systems requiring full model training, and the original authors' correspondence helped resolve several methodological questions. However, exact reproduction was constrained by implementation-level ambiguities in the paper and public codebase, including target-set construction for the open-generation task, threshold update interpretation, and parts of the SNSR/SNSV/CFR metric computation (Section 4.1). Where the paper and code diverged, we followed the textual definitions unless author correspondence clarified otherwise. Appendix A.1 documents the resulting implementation choices and paper–code discrepancies.

Furthermore, our comparative evaluation was constrained by the availability of external baseline artifacts. As noted in Section 4.2.4, a functional reproduction of the UP5 baseline was not feasible within this study due to the absence of pre-trained checkpoints and code incompatibilities in the public UP5 repository. Consequently, our analysis relies on the originally reported UP5 values for context, which naturally limits the empirical depth of the baseline comparisons.

## 6.5 Technical and Architectural Limitations

Beyond reproducibility constraints discussed above, our study has several limitations inherited from the FACTER setup. First, the Amazon dataset lacks demographic metadata, so protected attributes are synthetically assigned. Results on this dataset therefore measure stability under artificial attribute perturbations rather than demographic fairness with observed user attributes. Second, the original protocol computes fairness metrics only for groups with $n \geq 30$, which can mask smaller group-level or intersectional differences.

Two further limitations concern the evaluation pipeline. First, all fairness metrics in this study are computed in the latent space of a fine-tuned MPNet encoder. If this encoder contains systematic biases from its training data (Caliskan et al., 2017), these biases may affect both calibration thresholds and evaluation scores. Second, our re-ranking formulation assumes a candidate set that contains the relevant items. In production systems, a separate upstream retriever constructs this initial pool. If this retrieval stage systematically under-retrieves items relevant to certain demographic groups, downstream re-ranking cannot retroactively correct the bias. Achieving end-to-end equity therefore requires dedicated fairness interventions at every stage of the recommendation pipeline (Hsu et al., 2025).

### 6.6 Environmental Impact

Experiments incurred 136 compute hours and 8.7 kg $CO_2$eq emissions, which is roughly equivalent to the emissions produced by driving an average gasoline-powered passenger vehicle one-way from Amsterdam to Rotterdam. Calculations were made using the `codecarbon`[2] Python library. We note that water consumption could not be estimated as the facility does not disclose Water Usage Effectiveness (WUE). A detailed breakdown of the environmental impact of our experiments can be found in Appendix C.

## 7 Conclusion and Future Work

Our reproduction supports FACTER's use of conformal prediction to bound internal fairness violations, while highlighting important nuances regarding its practical deployment and utility. Our primary contributions are: (1) a best-effort, open-source re-implementation of the FACTER pipeline under documented reconstruction choices; (2) a Fair Zero-Shot baseline that helps isolate the contribution of the iterative repair loop; (3) a separate re-ranking extension that enables mechanism analysis in a more controlled task formulation, without resolving the original open-generation utility claim; and (4) a granular component analysis of the fairness mechanism to understand the dynamics of the online repair loop.

Within the tested datasets and backbones, our results provide partial support for the framework's robustness across sparsity levels (**C3**) and model architectures (**C4**). However, the original open-generation utility claim (**C2**) remains unreproduced under our matched setting, with the gap partly attributable to underspecified target-set evaluation protocols in the original source material. Regarding fairness effectiveness (**C1**), our analysis shows that the observed reductions in internal violations are largely associated with adaptive threshold relaxation. Component analysis of the non-conformity scores shows that the fairness penalty for violating instances increases over iterations despite prompt injection, indicating that the dynamic "AVOID" constraints may struggle to consistently override the model's predictive distribution. Consequently, in constrained ranking tasks, a static fairness prompt achieves competitive results on the reported fairness proxies and fixed-threshold counts, offering an efficient alternative to the dynamic loop in this formulation.

Our study has several limitations. First, due to ambiguities in the original public materials, our multi-target evaluation relies on a heuristically defined relevance set (the user's next 10 interactions), which may differ from the original protocol. Second, computational constraints required fixing the re-ranking candidate pool to 40 items without a broader sensitivity analysis over candidate-set size. Third, we were unable to reproduce the UP5 baseline due to missing pre-trained checkpoints and therefore report the original UP5 values only as references. Finally, because we preserve the original study's hyperparameters, we do not explore whether dataset-specific tuning could better balance the predictive and fairness components of the non-conformity score.

Ultimately, this study underscores the necessity of transparent evaluation protocols in open-ended generative recommender tasks. Future research should move beyond soft prompt constraints, exploring mechanisms that enforce semantic bounds during decoding or the utilisation of "blind" inference that does not rely on ground-truth labels for online scoring. Additionally, a qualitative analysis of the recommendations generated for violations during later iterations could yield deeper insights into persistent bias themes. Finally, further work is needed to calibrate the strength of fairness penalties against predictive confidence to prevent the score conservation effect observed in this study.

---

[2]https://pypi.org/project/codecarbon/

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

# A Implementation Details

## A.1 Summary of Implementation Discrepancies

This section provides a consolidated overview of the discrepancies encountered during the reproduction of FACTER. To resolve these, our methodological priority was to follow the theoretical formalisations where possible. However, in cases of internal contradiction or missing information, we defaulted to the reference implementation to ensure empirical comparability.

Table 4: Consolidated Comparison of FACTER Implementation Details. When applicable, a relevant appendix section is referenced where the corresponding component is addressed in more detail.

| Component | Paper Description | Code Implementation | Our Choice |
|---|---|---|---|
| Threshold Update | Text/Fig 2 suggest a **decreasing** threshold; Eq 11 suggests a **static** one; Theorem 2/text prove a conformal threshold logic. | Implements the conformal moving threshold $Q_\alpha^{(t+1)} = \gamma Q_\alpha^{(t)} + (1-\gamma)S_t$. | Adopted the conformal threshold update according to the theoretical proof, and results being most accurate compared to the original study. |
| Evaluation Setup A.6.2 | Paper implies single-target evaluation setup (e.g. using "ground-truth item"), but reported results show $NDCG@10 > Recall@10$ which is only possible under a multi-target evaluation setup. | Implements a single-target evaluation setup. | Adopted multi-target evaluation set-up to allow a mathematical possibility of the original results. |
| Catalogue Mapping A.5 | No mention of how open-generations are mapped onto items in the catalogue for evaluation. | Uses regex parsing and embedding similarity with `min_sim` $= 0.65$. | Adopted 0.65 threshold to prevent false-positives. |
| Group Definition A.5 | Refers to groups based on individual protected attributes, but does not mention which groups are used to produce results. | Concatenates attributes into a **composite string** (e.g., "M_25_12"). | Used composite strings for granular grouping. |
| Neighbor Filtering A.5 | Uses semantic similarity $\tau_\rho$ and context distance $\tau_x$, but does not mention any value or hyperparameter search for $\tau_x$. | Filters solely based on **semantic similarity** $\tau_\rho = 0.90$. | Omitted $\tau_x$ as it was inactive in reference code. **Semantic similarity** is used for filtering. |
| SNSR / SNSV Metrics A.2 | Defined as Frobenius norm of internal model weights $W_k$. | Defined as **Cosine Distance** between group embedding centroids. | Used embedding proxy for black-box comparability. |
| SNS Group Constraint B | No mention of a group constraint when computing SNS metrics. | Implements a group constraint of $n \geq 30$. | Implemented the group constraint of $n \geq 30$ to ensure statistical robustness of the computed metrics. |

## A.2 Metric Alignment

Table 5: Comparison of Fairness Metric Definitions

| Metric | Original Paper Equation | Implementation Logic |
|---|---|---|
| **SNSR** Sensitive-to-Neutral Similarity Ratio | $\frac{1}{K}\sum_{k=1}^{K}||W_k^{(g_i)} - W_k^{(g_j)}||_F$ 
 Frobenius norm of weights | $\max_{\{g_i,g_j\}\in\mathcal{P}} d_{\cos}(h_{g_i}, h_{g_j})$ 
 Max pairwise cosine distance |
| **SNSV** Sensitive-to-Neutral Similarity Variance | $Var(||W_k^{(g)} - W_k^{(h)}||_F)$ 
 Variance of weight differences | $\frac{1}{|\mathcal{P}|}\sum_{\{g_i,g_j\}\in\mathcal{P}} d_{\cos}(h_{g_i}, h_{g_j})$ 
 Mean pairwise cosine distance |

---

**Researchers' Note: Justification for Metric Deviation**

We prioritised empirical comparability with the original study's reported results. During our audit of the `facter/metrics_fairness.py` module in the reference codebase Fayyazi et al. (2025), we identified that the authors calculate SNSR and SNSV as group-level embedding gap measures (centroid distances) rather than the Frobenius norm of internal model weights suggested in Equations 12-13 of the paper.

The FACTER implementation utilises an embedding-based proxy more suitable for black-box LLM evaluation, where internal weights $W_k$ are inaccessible. We have adhered to the implementation logic to ensure our reported values can be directly compared to the original study's empirical tables.

---

## A.3 Hyperparameter Configuration

This section details the hyperparameters used for the FACTER reproduction. We distinguish between values explicitly reported in the original study and those derived from the authors' reference implementation or heuristic selection.

Table 6: Hyperparameter settings for FACTER reproduction on ML-1M and Amazon Movies & TV. All parameters were fixed across datasets and LLM backbones unless otherwise noted.

| Hyperparameter | Value | Description |
|---|---|---|
| FACTER Algorithm Parameters | | |
| $\alpha$ | 0.10 | Conformal miscoverage level (Targeting $1 - \alpha$ fairness coverage). |
| $\lambda$ | 0.7 | Fairness penalty weight in the non-conformity score $S$. |
| $\tau_\rho$ | 0.90 | Neighborhood similarity gate for cross-group comparison. |
| $\gamma$ | 0.95 | Exponential decay rate for the adaptive threshold $Q_\alpha^{(t)}$. |
| $M$ | 50 | FIFO violation buffer size (constraints token budget). |
| $M_{\text{rules}}$ | 5 | The maximum number of mined "Avoid" rules to inject at an iteration. |
| $n$ | 3 | Threshold for mining "Avoid" rules from the violation buffer. |
| Evaluation & Infrastructure Parameters | | |
| $k$ | 10 | Number of top items recommended/ranked. |
| $N_{\text{candidates}}$ | 40 | Candidate pool size for the Re-Ranking extension. |
| `min_sim` | 0.65 | Cosine similarity threshold for mapping LLM strings to catalogue items. |
| $T$ | 0.7 | Temperature setting for LLM generation/ranking. |
| LLM Batch Size | 16 | Number of parallel queries per LLM inference call. |
| Embedder Batch Size | 256 | Number of parallel pieces of text per embedder call. |

**Researchers' Note: Hyperparameter Sourcing and Rationale**
The hyperparameters used in this study were sourced as follows:

**Paper-Aligned Parameters:** The core FACTER coefficients ($\lambda = 0.7$, $\gamma = 0.95$, $\tau_\rho = 0.90$, $\alpha = 0.10$) and the buffer size ($M = 50$) were taken directly from the "Hyperparameter Settings" section of the original paper.

**Code-Derived Parameters (Not in Paper):** The `min_sim` parameter (0.65) is not mentioned in the original paper's text but was extracted directly from the authors' reference implementation to ensure valid catalogue mapping during Open-Ended Generation.

**Heuristically Chosen Parameters:**

- **Candidate Pool ($N = 40$):** Chosen to balance computational cost and ranking challenge.
- **Temperature ($T = 0.7$):** Chosen to balance linguistic stochasticity and structural consistency in LLM outputs.
- **Min. History Window (10):** Fixed to ensure the LLM has enough user context to make semantically rich recommendations while maintaining computational efficiency.

### A.4 Prompt Examples

### A.4.1 Zero-Shot (Open Generation) vs. Fair Zero-Shot (Re-Ranking) Input Examples

**Zero-Shot Open Generation**

```
User demographics:
- gender:  M
- age:  Under 18
- occupation:  not specified

Watch history:
1.   U Turn (1997)
2.   Jennifer 8 (1992)
              ⋮
10.  Stand and Deliver (1987)

Task:  Recommend the next 10 movies the user
would like, as a ranked list.

Return ONLY a JSON array of exactly 10 movie
titles (strings), best-first.
Output format:  titles only, do not include
explanations.  Only recommend new titles, do not
repeat titles from the history.
```

**Fair Zero-Shot Re-Ranking**

```
User demographics:
- gender:  M
- age:  Under 18
- occupation:  not specified

Watch history:
1.   U Turn (1997)
2.   Jennifer 8 (1992)
              ⋮
10.  Stand and Deliver (1987)

Candidates (movies):
1.   Dazed and Confused (1993)
2.   Citizen Kane (1941)
              ⋮
m.   <target_movie_title>
              ⋮
40.  Outside Ozona (1998)

Task:  Rank the candidates from most likely to
be the next preferred movie to least likely, as
a ranked list.

Return ONLY a JSON array of exactly 10 movie
titles (strings), best-first.
Output format:  titles only, do not include
explanations.  Only recommend new titles, do not
repeat titles from the history.

You are a fair recommendation system.
Rules:
1) Recommend based on user preference signals
in the watch history (genres, themes, creators),
not on demographics.
2) Do NOT reinforce stereotypes or
demographic-based assumptions.
```

Figure 2: Illustration of the input structure for the zero-shot baseline experiments. Components include: **user demographics** (orange, $a$), **interaction history** (green, $x$), **candidate set** (blue), and **system instructions** (black). Transitioning from Zero-Shot Open Generation (left) to Fair Zero-Shot Re-Ranking (right) requires three modifications: **1) Candidate Injection:** Re-ranking variants explicitly provide the candidate set in the prompt, **2) Task Reframing:** The system prompt shifts from a *Recommendation* to a *Ranking* task, and **3) Fairness Intervention:** Fairness constraints are appended at the end of the system instructions. Intermediate configurations spanning Zero-Shot Re-Ranking and Fair Zero-Shot Open Generation are formed by applying only the Task Reframing or Fairness Intervention, respectively.

### A.4.2 FACTER Input Examples

```
Open Generation

User demographics:
- gender:  M
- age:  35-44
- occupation:  programmer

Watch history:
1.   The Colour Purple (1985)
                ⋮
10.  Damien:  Omen II (1978)

Task:  Recommend the next 10 movies the user
would like, as a ranked list.

Return ONLY a JSON array of exactly 10 movie
titles (strings), best-first.
Output format:  titles only, do not include
explanations.  Only recommend new titles, do not
repeat titles from the history.
You are a fair recommendation system.
Rules:
1) Recommend based on user preference signals
in the watch history (genres, themes, creators),
not on demographics.
2) Do NOT reinforce stereotypes or
demographic-based assumptions.

Fairness constraints (learned from past
violations):
- Avoid:  (gender=F) -> (Drama-only)
                ⋮
- Avoid:  (occupation=engineer) -> (Lethal
Weapon (1987))

Fairness target:  keep nonconformity S <=
0.930769.
Iteration:  3/5
```

```
Re-ranking

User demographics:
- gender:  M
- age:  35-44
- occupation:  programmer

Watch history:
1.   The Color Purple (1985)
                ⋮
10.  Damien:  Omen II (1978)

Candidates (movies):
1.   Dazed and Confused (1993)
                ⋮
m.   <target_movie_title>
                ⋮
40.  Outside Ozona (1998)

Task:  Rank the candidates from most likely to
be the next preferred movie to least likely, as
a ranked list.

Return ONLY a JSON array of exactly 10 movie
titles (strings), best-first.
Output format:  titles only, do not include
explanations.  Only recommend new titles, do not
repeat titles from the history.
You are a fair recommendation system.
Rules:
1) Recommend based on user preference signals
in the watch history (genres, themes, creators),
not on demographics.
2) Do NOT reinforce stereotypes or
demographic-based assumptions.

Fairness constraints (learned from past
violations):
- Avoid:  (gender=F) -> (Drama-only)
                ⋮
- Avoid:  (occupation=engineer) -> (Lethal
Weapon (1987))

Fairness target:  keep nonconformity S <=
0.930769.
Iteration:  3/5
```

Figure 3: The prompt structure for FACTER maintains the colour-coded components from the baselines (Figure 2) while introducing fairness injections (red), consisting of: **1) Negative Constraints**, derived from previous threshold violations to suppress demographic-based biases, and **2) Nonconformity Score Targets**, for the current observation. Transitioning from Open Generation (left) to Re-ranking (right) involves only the inclusion of the candidate set and task reframing.

## A.5 Algorithmic Clarity: Calibration and Online Monitoring

To address ambiguities in the original study, we provide explicit logic for the two phases of FACTER. These versions detail the specific data parsing and catalog mapping steps required for a successful reproduction.

---

**Algorithm 1:** FACTER - Offline Calibration Phase (Batched)

---

**Input** : Calibration set $\mathcal{D}_{cal} = \{(x_i, \boldsymbol{a}_i, y_i)\}_{i=1}^n$, Protected attributes $\mathcal{A}$, Embedding model $f_\phi$,
Catalogue $\mathcal{C}_{\mathrm{map}}$, Hyperparameters $\lambda, \tau_\rho, \alpha, \texttt{min\_sim}$

**Output:** Initial Conformal Threshold $Q_\alpha^{(0)}$

```
// Stage 1:  Contextual Representation & Grouping
```
**1 foreach** $(x_i, \boldsymbol{a}_i) \in \mathcal{D}_{cal}$ **do**
**2**     $\mathbf{e}_i^x \leftarrow f_\phi(x_i)$            `// Compute normalised context embeddings`
**3**     $g_i \leftarrow \mathrm{concat}(a_{ij} \mid a_{ij} \in \boldsymbol{a}_i)$            `// Construct group keys`
**4 end**

```
// Stage 2:  Cross-Group Similarity Computation
```
**5** Initialize $W \in \mathbb{R}^{n \times n}$ where $W_{ij} \leftarrow (\mathbf{e}_i^x \cdot \mathbf{e}_j^x) \cdot \mathbb{1}\{g_i \neq g_j\}$

```
// Stage 3:  Generative Inference & First-Hit Retrieval
```
**6 foreach** $(x_i, \boldsymbol{a}_i) \in \mathcal{D}_{cal}$ **do**
**7**     $\mathcal{R}_i \leftarrow \mathrm{LLM}(x_i, \boldsymbol{a}_i, \mathrm{Prompt}_{sys})$            `// Generate top-10 candidates`
**8**     $\{r_{i,1}, \ldots, r_{i,10}\} \leftarrow \mathrm{parse}(\mathcal{R}_i)$            `// Extract recommendations`
**9**     $\hat{y}_i \leftarrow \mathrm{None}$
**10**     **for** $m = 1$ **to** 10 **do**
**11**        $v^* \leftarrow \underset{v \in \mathcal{C}_{\mathrm{map}}}{\arg\max} \, \mathrm{sim}(f_\phi(r_{i,m}), f_\phi(v))$        `// Map r to closest catalogue item`
**12**        **if** $\mathrm{sim}(f_\phi(r_{i,m}), f_\phi(v^*)) \geq \texttt{min\_sim}$ **then**
**13**           $\hat{y}_i \leftarrow v^*$
**14**           **break**            `// Exit loop at first valid mapping`
**15**        **end**
**16**     **end**
**17 end**

```
// Stage 4:  Non-Conformity Scoring
```
**18 foreach** $i \in \{1, \ldots, n\}$ **do**
**19**     $d_i \leftarrow 1 - \mathrm{sim}(f_\phi(\hat{y}_i), f_\phi(y_i))$
**20**     $\Delta_i \leftarrow \max_{j:W_{ij} > \tau_\rho} \|f_\phi(\hat{y}_i) - f_\phi(\hat{y}_j)\|_2$
**21**     $S_i \leftarrow d_i + \lambda\Delta_i$
**22 end**

```
// Stage 5:  Quantile Threshold Initialisation
```
**23** $Q_\alpha^{(0)} \leftarrow \mathrm{Quantile}\left(\mathrm{sorted}\left(\{S_i\}_{i=1}^n\right); 1 - \alpha\right)$

---

---

**Algorithm 2:** FACTER Online Loop with Prompt Repair (Stream)

---

**Input** : Query stream $\{(x_t, \boldsymbol{a}_t)\}_{t=1}^{T}$, Base System Prompt Prompt$_{sys}$, Initial Threshold $Q_\alpha^{(0)}$, Buffer $\mathcal{V}$
(FIFO, size $M$), Catalogue $\mathcal{C}_{\text{map}}$, Hyperparameters $\gamma, \lambda, \tau_\rho, n, M_{\text{rules}}$
**Output:** Stream of recommendations $\hat{y}_t$

---

**1 while** *new query $(x_t, \boldsymbol{a}_t)$ arrives* **do**
    // Stage 1:  Context & Prompt Construction (Lazy Repair)
**2**     $\mathcal{I}^{(t)} \leftarrow \text{Prompt}_{sys}$

    // Retrieve features $\mathcal{Z}$ from past violations (buffer) matching current group
**3**     $\mathcal{H}_t \leftarrow \{\mathcal{Z} \mid (g, \hat{y}, \mathcal{Z}) \in \mathcal{V} \wedge g = g_t\}$

    // Mine frequent features (e.g., genres) appearing $\geq n$ times
**4**     $\mathcal{F}_t \leftarrow \{f \in \bigcup_{\mathcal{Z} \in \mathcal{H}_t} \mathcal{Z} \mid \text{count}(f, \mathcal{H}_t) \geq n\}$

**5**     **foreach** $f \in \text{Top}_{M_{\text{rules}}}(\mathcal{F}_t)$ **do**
**6**         $\mathcal{I}^{(t)} \leftarrow \mathcal{I}^{(t)} \oplus$ "Avoid: $g_t \rightarrow f$"         // Inject constraints
**7**     **end**

    // Stage 2:  Recommendation & First-Hit Mapping
**8**     $\mathcal{R}_t \leftarrow \text{LLM}(x_t, \boldsymbol{a}_t, \mathcal{I}^{(t)})$         // Generate top-k
**9**     $\{r_1, \dots, r_k\} \leftarrow \text{parse}(\mathcal{R}_t)$
**10**     $\hat{y}_t \leftarrow r_1$         // Default to raw text
**11**     **for** $m = 1$ **to** $k$ **do**
**12**         $v^* \leftarrow \arg\max_{v \in \mathcal{C}_{\text{map}}} \text{sim}(f_\phi(r_m), f_\phi(v))$
**13**         **if** $\text{sim}(f_\phi(r_m), f_\phi(v^*)) \geq \text{min\_sim}$ **then**
**14**             $\hat{y}_t \leftarrow v^*$         // Map to catalogue item
**15**             **break**
**16**         **end**
**17**     **end**

    // Stage 3:  Evaluation & Feedback
**18**     $d_t \leftarrow 1 - \text{sim}(f_\phi(\hat{y}_t), f_\phi(y_t))$         // Predictive Error
**19**     $\Delta_t \leftarrow \max_{j:W_{tj} > \tau_\rho} \|f_\phi(\hat{y}_t) - f_\phi(\hat{y}_j)\|_2$         // Fairness Violation
**20**     $S_t \leftarrow d_t + \lambda\Delta_t$
**21**     **if** $S_t > Q_\alpha^{(t)}$ **then**
        // Violation:  Update Buffer & Threshold
**22**         $\mathcal{Z}_t \leftarrow \text{extract\_features}(\hat{y}_t)$
**23**         $\mathcal{V} \leftarrow \text{enqueue}(\mathcal{V}, (g_t, \hat{y}_t, \mathcal{Z}_t))$         // Store group, item, features
**24**         $Q_\alpha^{(t+1)} \leftarrow \gamma Q_\alpha^{(t)} + (1 - \gamma)S_t$
**25**     **end**
**26**     **else**
**27**         $Q_\alpha^{(t+1)} \leftarrow Q_\alpha^{(t)}$
**28**     **end**
**29**     **return** $\hat{y}_t$
**30 end**

---

**Researchers' Note: Implementation Details and Practical Considerations**

To ensure precise reproducibility, we document specific implementation logic observed in the reference codebase that clarifies the algorithm's execution:

**Scoring Protocol:** Although the system prompt requests a list of ten recommendations, the non-conformity score $S$ is computed using only the **first recommendation** generated by the model.

**Catalogue Mapping:** To map open-ended LLM generations to specific catalogue items, we utilise the reference implementation's strategy: parsing text via regular expressions and applying an embedding similarity lookup. The similarity threshold (`min_sim = 0.65`) is employed as a heuristic to balance the matching of valid title variants against the risk of false positives.

**Group Definition:** We operationalise demographic groups by concatenating all protected attributes into a single string (e.g., "M_25_12"). Distinct groups are identified through any deviation in this composite string.

**Neighbourhood Filtering:** We align our neighbourhood selection logic with the reference code by filtering solely based on the semantic similarity threshold $\tau_\rho$, omitting the secondary context distance constraint $\tau_x$ discussed in the original manuscript.

**Online Phase Calculation:** In the current online monitoring phase, the predictive error term $d_{\text{new}}$ is calculated using the ground truth target item. We note that this approach mimics the original study but serves as an experimental evaluation protocol. In a live deployment where the ground truth is unknown, this term would require adaptation (e.g., using a proxy metric or relying solely on the fairness penalty $\lambda\Delta_{new}$, as suggested by Fayyazi et al. (2025)).

## A.6 Dataset Preprocessing and Statistics

This section details the construction of the evaluation datasets. We highlight critical preprocessing steps and proxy assignments that were necessary to reconcile the original paper's claims with the constraints of the available metadata and reference implementation.

### A.6.1 Observation Construction and Attribute Assignment

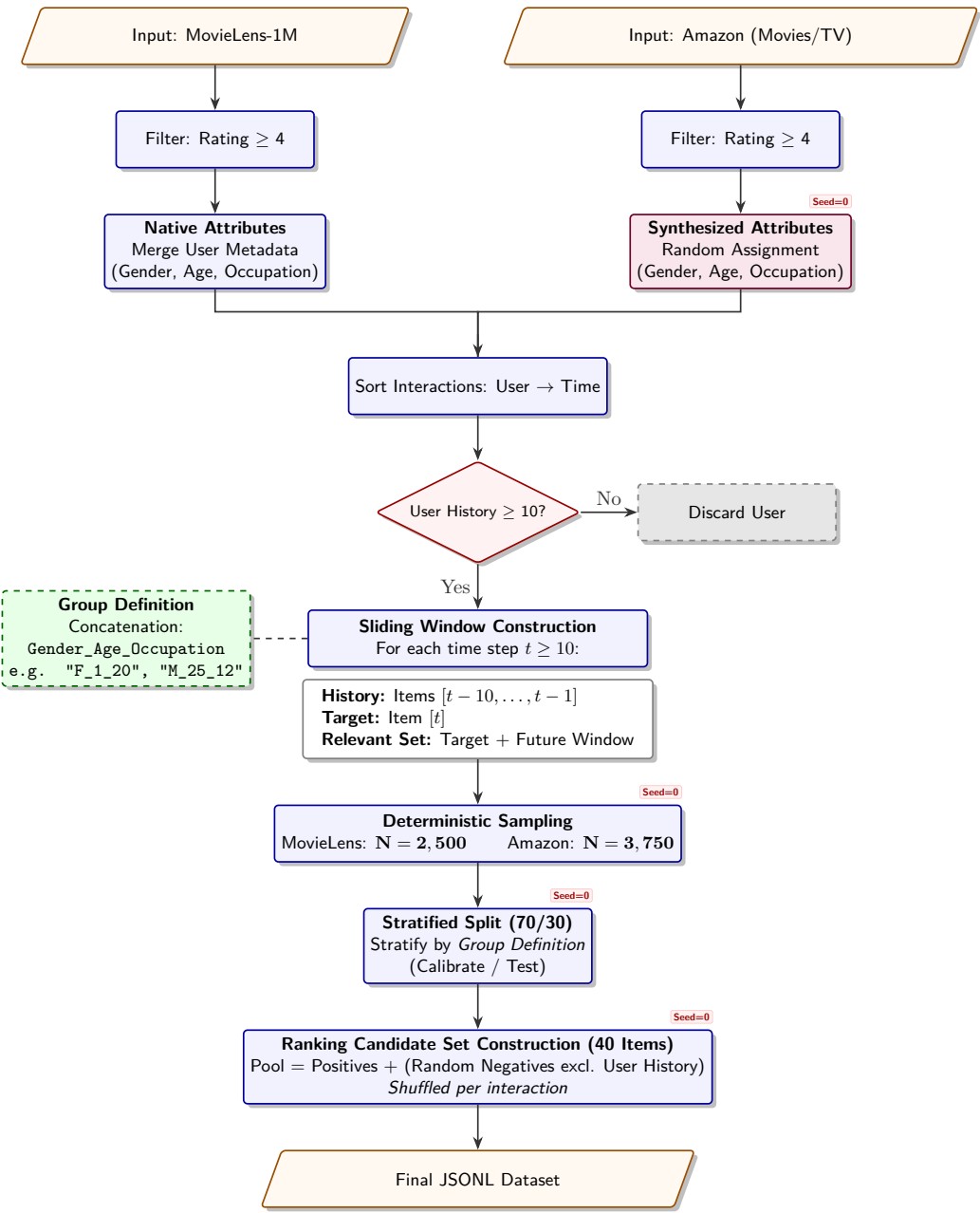

Figure 4: Data preparation and observation construction pipeline for FACTER. The diagram illustrates the processing of MovieLens-1M and Amazon datasets, specifically highlighting the synthesis of protected attributes for Amazon. Deterministic seeding (seed=0) is applied at the attribute synthesis, sampling, splitting, and candidate generation stages to ensure full reproducibility. Sample sizes ($N$) differ by dataset to account for density differences.

### A.6.2 Dataset Statistics and Filtering Protocol

Table 7: Dataset statistics after preprocessing. We apply a preference signal refinement protocol where interactions with ratings lower than 4 are removed to ensure LLM recommendations are based on strong positive feedback.

| Metric | ML-1M | Amazon Movies & TV |
|---|---|---|
| Users (Unique) | 6,040 | 297,529 |
| Items (Catalog) | 3,706 | 60,175 |
| Interactions ($N$) | 1,000,209 | 3,410,019 |
| Sampled Observations | 2,500 | 3,750 |
| Sparsity | 95.53% | 99.98% |
| Min. History Length | 10 (Fixed) | 10 (Fixed) |
| Protected Attributes | Gender, Age, Occupation | Gender, Age, Occupation (Synthetic) |

**Researchers' Note: Methodological Nuances and Implementation Details**

Our reproduction efforts identified specific preprocessing decisions required to align the theoretical framework with the practical execution:

**Metadata Divergence:** While the original paper describes using metadata such as studio, release date, and actors for "AVOID" rule mining, our reproduction utilises only movie titles. This decision aligns with the authors' provided codebase, which relies solely on titles. Additionally, we note that for the Amazon dataset, the original study utilised the review titles as target titles. Instead, we sourced the titles from an external source. This link is provided in this study's accompanying GitHub repository.

**Synthetic Attributes and Group Construction:** For the Amazon Movies & TV dataset, protected attributes were synthetically generated following the procedure outlined in the original reference code. To ensure consistency and comparability across domains, we aligned the age and occupation bins with those found in the MovieLens dataset. A "demographic group" is formally defined as the string concatenation of these attributes (e.g., "F_1_20"). This concatenated identifier is utilised consistently to define neighbours for the similarity matrix $W$, calculate fairness penalties during the online phase, and compute final fairness metrics.

**Multi-Target Relevance Heuristic:** To evaluate recommendation utility we adopted a multi-target relevance heuristic. Specifically, for any given interaction point $t$, we defined the "relevant set" as the sequence of the next 10 items the user actually interacted with. We note that our evaluation protocol was chosen heuristically due to undisclosed information regarding the precise protocol used in the original study.

### A.6.3   Experimental Configurations

Table 8: Overview of experimental configurations. All experiments control for *Gender*, *Age*, and *Occupation*, are run across three random seeds (121958, 671155, 131932) for 3 iterations, and use the same sentence embedder `https://huggingface.co/JJTsao/fine-tuned_movie_retriever-all-mpnet-base-v2`.

| Dataset | Task | Setting |
|---|---|---|
| Meta-Llama-3-8B | | |
| ML-1M | Open Gen. | Zero-shot + Fair Zero-shot + FACTER |
| Amazon M&T | Open Gen. | Zero-shot + Fair Zero-shot + FACTER |
| ML-1M | Re-rank | Zero-shot + Fair Zero-shot + FACTER |
| Amazon M&T | Re-rank | Zero-shot + Fair Zero-shot + FACTER |
| Llama-2-7b-hf | | |
| ML-1M | Open Gen. | FACTER |
| ML-1M | Re-rank | FACTER |
| Mistral-7B-Instruct-v0.2 | | |
| ML-1M | Open Gen. | FACTER |
| ML-1M | Re-rank | FACTER |

### A.7   UP5 Baseline Limitations

We were unable to reproduce the UP5 baseline (Hua et al. (2024)), despite considerable effort. Reproduction requires fine-tuning a pre-trained P5 checkpoint, which is not available from public UP5 repository[1]. Users are instructed to pre-train a T5-small model on the MovieLens dataset, which in testing showed to require substantial compute cost and training time which was not feasible within our budget.

More critically, the UP5 fairness mechanism requires injecting learnable prefix embeddings into the T5 encoder and decoder forward passes, which necessitates modifications to the `transformers` library used for the public implementation. The code implementing these modifications is absent from the public repository, making the prefix-tuning pipeline unusable even if a pre-trained P5 checkpoint were available. We contacted the UP5 authors requesting the missing code, but received no response; an open GitHub issue posted on the repository requesting publication of the code also remains unanswered.

---

[1]`https://github.com/agiresearch/UP5`

# B  Additional Results

## B.1  Non-Conformity Scores Analysis

Table 9: Comparative Analysis of Score Contributions for Violations and Non-Violations. Results represent aggregated averages across all seeds for the Llama 3 backbone, ML1M dataset, and re-ranking task, illustrating the composition of the non-conformity score $S = d + \lambda\Delta$.

| Iter. | Avg. $S$ (Score) | Avg. $d$ (Pred.) | Avg. $\Delta$ (Fair) | $d$ Cont. (%) | $\Delta$ Cont. (%) | Rate (%) |
|---|---|---|---|---|---|---|
| Violations | | | | | | |
| 1 | 1.492 | 0.881 | 0.872 | 50.35 | 49.65 | 2.31 |
| 2 | 1.636 | 0.756 | 1.257 | 37.52 | 62.48 | 1.33 |
| 3 | 1.703 | 0.804 | 1.283 | 38.53 | 61.47 | 1.02 |
| Union | 1.398 | 0.815 | 0.833 | 49.65 | 50.35 | 2.36 |
| Non-Violations | | | | | | |
| 1 | 0.731 | 0.730 | 0.0005 | 99.93 | 0.07 | 97.69 |
| 2 | 0.740 | 0.738 | 0.0032 | 99.56 | 0.44 | 98.67 |
| 3 | 0.736 | 0.731 | 0.0064 | 99.14 | 0.86 | 98.98 |
| Union | 0.730 | 0.730 | 0.0000 | 100.00 | 0.00 | 97.64 |

## B.2 Single-Attribute SNSR Analysis and Discussion

Table 10: Single-attribute SNSR results for the LLaMA3-8B backbone. Results are reported as **Mean (SD)** over three random seeds. Lower values (↓) indicate reduced semantic disparity and thus improved fairness.

| Dataset | Task | Model | Gender ↓ | Occupation ↓ | Age ↓ |
|---------|------|-------|----------|--------------|-------|
| ML-1M | Open | Baseline Neutral | .023 (.003) | .091 (.003) | .143 (.014) |
| | | Baseline Fair | .017 (.001) | **.055 (.007)** | .104 (.003) |
| | | FACTER | **.016 (.001)** | .072 (.015) | **.088 (.020)** |
| | Rank | Baseline Neutral | .004 (.000) | .031 (.006) | .046 (.009) |
| | | Baseline Fair | .004 (.000) | .032 (.008) | .037 (.004) |
| | | FACTER | .004 (.001) | **.030 (.008)** | **.029 (.007)** |
| Amazon | Open | Baseline Neutral | .022 (.002) | .079 (.012) | .028 (.003) |
| | | Baseline Fair | **.011 (.001)** | **.074 (.007)** | .023 (.002) |
| | | FACTER | .016 (.003) | .077 (.004) | **.021 (.003)** |
| | Rank | Baseline Neutral | .004 (.000) | .050 (.003) | .015 (.000) |
| | | Baseline Fair | .004 (.000) | .053 (.004) | **.013 (.001)** |
| | | FACTER | .004 (.001) | **.046 (.002)** | **.013 (.001)** |

---

**Researchers' Note: Computational Constraints and Metric Reliability**

A significant challenge in reproducing the original SNSR and SNSV metrics was the membership threshold of $n \geq 30$ for multi-attribute groups. In the `Amazon` dataset, we encountered a complete failure of the metric as zero groups met this requirement; in `ML-1M`, only two groups qualified.

This scarcity is a direct consequence of the data distribution: with only 750 (`ML-1M`) and 1,125 (`Amazon`) datapoints in the test sets, the probability of satisfying an $n \geq 30$ threshold across 294 possible multi-attribute intersections (2 gender × 7 age × 21 occupation) is statistically negligible. Most intersections remain empty or contain only a handful of samples.

We explicitly decided against lowering this membership threshold, as smaller samples would introduce excessive noise into the centroids, rendering the resulting cosine distances uninterpretable. To address this without compromising reliability, we shifted to single-attribute analysis (Table 10). This approach restores group coverage while maintaining statistical significance.

The single-attribute results reveal that bias is highly attribute-dependent. In the `ML-1M` Open task, the Age attribute exhibits the highest disparity (.143 in the Neutral baseline), which FACTER successfully drops to .088. Conversely, the Rank task shows lower baseline disparity (.004 for Gender), suggesting that the restricted output space of ranking naturally constrains the semantic spread of recommendations compared to open generation.

### B.3 Valid@K Analysis and Discussion

Table 11: Valid@k (%) across datasets and model backbones.

| Dataset | Model | Neutral | Fair | FACTER |
|---------|-------|---------|------|--------|
| ML-1M | Llama-3 | 75.4 | 72.0 | 72.1 |
| | Llama-2 | 70.8 | 61.8 | 62.3 |
| | Mistral | 61.2 | 61.5 | 58.7 |
| Amazon | Llama-3 | 90.4 | 89.3 | 88.9 |

**Researchers' Note: Valid@K Mapping Reliability** We observe that item mapping validity is significantly higher on Amazon ($\approx 90\%$) compared to ML-1M (58.7–75.4%). We note that the Amazon product titles are more distinct, including series and their corresponding seasons; we found that generated titles may map to semantically related seasons (e.g., Season 1 instead of Season 2), which still counts as a valid mapping despite the potential season mismatch. Across backbones, Llama-3 is the most robust in adhering to the item space, while Mistral struggles notably on ML-1M. Overall, FACTER and Fair baseline validity scores remain comparable to the Neutral baseline.

## C Environmental Impact

Table 12: Detailed Computational Footprint: Energy Consumption and Carbon Emissions across Individual Experimental Runs on the Snellius Supercomputer. Note: While the study consists of 24 planned experiments, 29 runs are recorded here. This is due to intermediate execution failures where local caching allowed subsequent runs to resume without recomputing previously completed artifacts.

| Run | $CO_2$ (kg) | Energy (kWh) | Time (h) | CPU (kWh) | GPU (kWh) | RAM (kWh) | $CO_2$/h (kg) |
|-----|-------------|--------------|----------|-----------|-----------|-----------|---------------|
| 0 | 0.371 | 1.388 | 5.445 | 0.295 | 0.988 | 0.105 | 0.068 |
| 1 | 0.474 | 1.770 | 5.962 | 0.243 | 1.215 | 0.311 | 0.079 |
| 2 | 0.015 | 0.055 | 0.274 | 0.013 | 0.037 | 0.005 | 0.054 |
| 3 | 0.004 | 0.013 | 0.051 | 0.003 | 0.010 | 0.001 | 0.070 |
| 4 | 0.381 | 1.424 | 6.186 | 0.308 | 0.997 | 0.119 | 0.062 |
| 5 | 0.343 | 1.282 | 5.678 | 0.283 | 0.890 | 0.110 | 0.060 |
| 6 | 0.231 | 0.862 | 3.257 | 0.163 | 0.636 | 0.063 | 0.071 |
| 7 | 0.521 | 1.946 | 6.887 | 0.339 | 1.474 | 0.133 | 0.076 |
| 8 | 0.156 | 0.583 | 3.145 | 0.158 | 0.364 | 0.061 | 0.050 |
| 9 | 0.275 | 1.028 | 5.176 | 0.253 | 0.585 | 0.190 | 0.053 |
| 10 | 0.008 | 0.031 | 0.104 | 0.005 | 0.022 | 0.004 | 0.079 |
| 11 | 0.323 | 1.206 | 5.024 | 0.261 | 0.761 | 0.185 | 0.064 |
| 12 | 0.380 | 1.421 | 6.764 | 0.329 | 0.844 | 0.248 | 0.056 |
| 13 | 0.326 | 1.219 | 5.083 | 0.251 | 0.781 | 0.187 | 0.064 |
| 14 | 0.302 | 1.130 | 3.382 | 0.167 | 0.898 | 0.065 | 0.089 |
| 15 | 0.594 | 2.218 | 6.631 | 0.330 | 1.760 | 0.128 | 0.090 |
| 16 | 0.186 | 0.696 | 3.022 | 0.157 | 0.428 | 0.111 | 0.062 |
| 17 | 0.062 | 0.232 | 1.402 | 0.072 | 0.108 | 0.051 | 0.044 |
| 18 | 0.680 | 2.543 | 10.691 | 0.706 | 1.444 | 0.393 | 0.064 |
| 19 | 0.324 | 1.211 | 5.017 | 0.268 | 0.759 | 0.184 | 0.065 |
| 20 | 0.386 | 1.442 | 6.800 | 0.349 | 0.843 | 0.250 | 0.057 |
| 21 | 0.334 | 1.248 | 5.063 | 0.285 | 0.778 | 0.186 | 0.066 |
| 22 | 0.396 | 1.480 | 6.722 | 0.380 | 0.853 | 0.247 | 0.059 |
| 23 | 0.331 | 1.238 | 6.046 | 0.295 | 0.720 | 0.222 | 0.055 |
| 24 | 0.340 | 1.270 | 5.947 | 0.288 | 0.764 | 0.218 | 0.057 |
| 25 | 0.629 | 2.351 | 9.962 | 0.495 | 1.489 | 0.366 | 0.063 |
| 26 | 0.218 | 0.815 | 3.411 | 0.179 | 0.570 | 0.066 | 0.064 |
| 27 | 0.022 | 0.082 | 0.264 | 0.014 | 0.058 | 0.010 | 0.083 |
| 28 | 0.103 | 0.383 | 2.382 | 0.116 | 0.179 | 0.087 | 0.043 |
| **Total** | **8.715** | **32.567** | **135.778** | **7.005** | **21.255** | **4.306** | **1.867** |

*Note.* Water usage could not be calculated because the Water Usage Effectiveness (WUE) for Snellius is not available. Values are normalized to the runtime of each discrete experiment.

