# OpenReview forum: "Reproducing FACTER: Fairness via Conformal Thresholding and Prompt Repair"
_TMLR — Accepted by TMLR_

### Review · Reviewer_tXWG · 2026-03-08

**Summary Of Contributions:**

This submission presents a reproduction study of FACTER, a prior method for fairness in LLM-based recommendation. The paper re-implements the original pipeline, evaluates the main empirical claims on MovieLens-1M and Amazon, adds a constrained re-ranking extension, and performs component analysis on the thresholding and prompt-repair mechanism. I think this is careful and useful work in a limited sense, but the contribution is still quite small. The paper does not introduce a new method, a new theory, or a new benchmark. Its main value comes from reproduction effort and follow-up analysis

**Audience:**

Yes

**Audience Explanation:**

I think some readers in the TMLR audience would be interested in this paper because reproducibility studies in fairness and LLM-based recommendation are worth having, especially when the original method is black-box, training-free, and practically motivated. That said, I do not think the level of insight here is strong enough to make this an especially impactful journal contribution. The paper is mostly useful as a careful check on a previous result, not as a paper that substantially changes how people should think about the problem.

**Broader Impact Concerns:**

I do not see any notable broader social or environmental concern beyond the ordinary cost of running experiments, and the paper already reports its compute and emissions clearly.

**Claims And Evidence:**

No

**Claims Explanation:**

The evidence only partially supports the paper’s claims. The paper does show that some high-level trends of FACTER can be reproduced, especially the adaptive threshold behavior and the consistency across several model backbones. However, the central conclusions are weakened by substantial implementation ambiguity in the original work, reliance on private correspondence with the original authors, and the fact that an important baseline, UP5, could not be reproduced. In addition, the strongest discrepancy in the paper, namely the large gap in open-ended utility, is explicitly attributed by the authors to evaluation-protocol sensitivity rather than a clear failure of the original method. This makes the paper more suggestive than conclusive. It raises doubts about the robustness of the original claims, but it does not cleanly establish that the original paper was wrong.

**Requested Changes:**

The paper would be stronger if it more clearly separated what is a faithful reproduction from what is a new task reformulation introduced by the authors. Right now, the re-ranking extension is practical, but it also changes the problem setting in an important way, so it should not be used to indirectly rescue the original utility claim. The paper should also make the evaluation mismatch more explicit and document exactly which details were unavailable from the public artifacts, because this limitation is central to the interpretation of the results. Finally, the paper should be more careful in stating what it has actually shown: it provides evidence that FACTER’s reported gains are sensitive to implementation and evaluation details, and that the adaptive threshold may explain much of the apparent improvement, but it does not fully settle whether the original claims hold under a truly matched implementation.

---

> ### Author Response · Authors · 2026-04-10
> **Reframing Scope, Claims, and Reproducibility Findings**
>
> Thank you for taking the time to provide us with your constructive feedback. We hear your concern regarding the novelty of insights in our paper. Our aim, in light of this being a reproducibility study, is to add value by investigating the reproducibility and generalizability of the original work while providing a wider perspective on the operational boundaries of the framework.
>
> Your feedback gave us insights into the shortcomings of our framing, particularly regarding the distinction between reproduction and extension. Concretely, we have made the following changes based on your feedback:
>
> **1: Separation of Reproduction and Task Reformulation.** We agree that the line between reproducing the original work and our proposed extension was previously ambiguous. We have made adjustments in the Abstract, Introduction, Claims, Results, and Conclusion to explicitly state that the re-ranking task is an extension meant to investigate the method in a constrained setting. We clearly distinguish that this extension was not intended to "rescue" or reconcile the original open-generation utility results, which we treat as a separate investigative outcome.
>
> **2: Documentation of Implementation Ambiguities.** We realize that missing or inconsistent details were previously scattered throughout the paper, making it difficult to find a clear overview of the discrepancies impacting reproducibility. We have gathered the most important inconsistencies and our corresponding implementation choices into a comprehensive "Investigative Audit" table in Appendix B. We now refer to this appendix in relevant sections of the main body to provide a central registry of the hurdles encountered during reproduction.
>
> **3: Reframing of Claims and Conclusions.** We have adjusted our conclusions in the Results and Discussion sections to be more factual regarding what our results establish. While we show that results are highly sensitive to implementation and evaluation details, we have tempered our claims to acknowledge that undisclosed protocols in the original work may explain the observed gaps. We focus on the finding that FACTER’s apparent gains are largely driven by adaptive threshold relaxation.
>
> **4: Addressing the UP5 Baseline and Methodology.** We address the missing UP5 baseline and the corresponding missing artifacts in Appendix B.5. We want to clarify that our claims are not dependent on a comparison against this external baseline; instead, we focus on comparisons against zero-shot baselines to isolate the impact of FACTER’s specific adaptive mechanisms. We now state this clarification clearly in section 4.2.4, where we mention that UP5 is not reproduced. By doing so, we believe the missing baseline does not fundamentally weaken the framing of our core findings.

---

### Review · Reviewer_7rM9 · 2026-03-24

**Summary Of Contributions:**

This paper presents a careful reproduction study of FACTER for fair LLM-based recommendation and adds a re-ranking extension to better understand the method in a constrained setting. The main finding is that the original open-ended setting does not reproduce the reported utility, while the re-ranking setup yields more reasonable utility and allows a clearer analysis of the fairness mechanism. The most interesting conclusion is that much of FACTER’s apparent gain seems to come from adaptive threshold relaxation, and that a static fairness prompt can be competitive with the full repair loop in the constrained setting.

**Audience:**

Yes

**Audience Explanation:**

Yes—some TMLR readers would be interested because the paper speaks to reproducibility, fairness, and evaluation reliability in LLM-based recommenders, all of which are relevant to trustworthy ML research. Its findings matter especially because it questions whether prior fairness gains are robust, shows that results depend strongly on evaluation setup, and proposes a simpler constrained re-ranking alternative with similar fairness benefits.

**Claims And Evidence:**

Yes

**Claims Explanation:**

The evidence is partially convincing but not fully definitive. The paper does a good job of testing its claims across datasets, models, and ablations, and it is transparent about failures to reproduce some original results. However, the strongest fairness improvements mainly appear with the authors’ adaptive thresholding or added re-ranking method, while the original utility results do not reproduce well and some fairness metrics are inconsistent or unavailable. So overall, the support is clear and careful, but it only backs a qualified version of the claims.

**Requested Changes:**

1. The largest issue is the unreconciled utility gap (roughly 0.03 vs. 0.44 NDCG@10), which the paper itself attributes to underspecified evaluation details, especially how the relevant target set is defined. This needs to be fully specified and, ideally, reconciled experimentally.
2. The reported fairness gains seem to come mainly from the adaptive threshold becoming more permissive, while fixed-threshold violations and external fairness metrics remain close to baselines or sometimes worsen. That makes the current framing too strong.
3. Key details such as the multi-target setup, threshold dynamics, and metric definitions appear insufficiently specified in the public materials. For a reproducibility-focused paper, that level of dependence on missing details is a major weakness.
4. The strongest results come from the added constrained re-ranking setup, not from strict reproduction of the original open-generation setting. The paper should make that distinction much more explicit in its claims and conclusions.

---

> ### Author Response · Authors · 2026-04-10
> **Separating Reproduction from Re-ranking Extension**
>
> Thank you for your constructive feedback and for recognizing the transparency of our reproduction efforts. We appreciate the opportunity to clarify the scope of our findings and the specific implementation hurdles encountered during the study.
> Based on your suggestions and requested changes, we have implemented the following adjustments:
>
> **1: Reconciling the Utility Gap.** Regarding the utility disparity, we believe that reconciliation is not possible under a strict reproduction of the open-ended generation task without significantly altering the original evaluation protocol. We reproduced the experimental set-up considering all publicly available artifacts. This interpretation is also supported by prior work in generative recommendation, where unconstrained generation over the full item space is often avoided by introducing candidate generation, candidate-constrained prompting, or retrieval/search stages. For example, Zero-Shot Next-Item Recommendation uses an external module to generate candidate items before prompting the LLM (Wang and Lim, 2023), P5 includes next-item prompts over explicit candidate sets (Geng et al., 2022), and GPT4Rec formulates recommendation as query generation followed by search over the catalog (Li et al., 2023). More broadly, the survey of Li et al. (2024) explicitly contrasts direct generation from the complete item pool with conventional multi-stage pipelines that include stages such as score computation and re-ranking. We now clarify in Section 6.1 that near-zero scores under a strict open-generation-and-matching protocol are not inherently anomalous, and that our separate re-ranking extension should be understood as a standard constrained formulation rather than an ad hoc rescue of the original utility claim.
>
> **2: Reframing Claims and Fairness Gains.** We have reframed our main claims in the Results and Discussion sections to be more factual. We now more explicitly characterize FACTER as a risk management system where violation reductions are primarily driven by adaptive threshold relaxation, rather than a fundamental correction of model bias. We acknowledge that external fairness metrics (CFR, SNSR) often showed no improvement or even slight degradation compared to simple static baselines, which is discussed in section 6.2 of our paper.
>
> **3: Documentation of Missing Details.** We agree that the dependence on missing details is a challenge; however, in the context of a reproducibility study, this is a primary finding. We have consolidated all identified ambiguities and our corresponding implementation choices, including the multi-target setup, threshold dynamics, and metric definitions, into a clear overview in Appendix B. While our results show FACTER is sensitive to these implementation choices, we have adjusted the wording of our contribution that this is a best-effort reproduction audit rather than a definitive adjudication of the original results.
>
> **4: Separation of Reproduction and Task Reformulation.** We have restructured the Abstract, Introduction, Claims, Results, and Conclusion to clearly separate the strict reproduction of the original open-generation setting from our proposed re-ranking extension. We explicitly state that the high utility magnitudes are achieved through this task reformulation and should not be used to indirectly validate the original open-ended generation utility claims.
>
> References:
> Lei Wang and Ee-Peng Lim. Zero-shot next-item recommendation using large pretrained language models, 2023. URL https://arxiv.org/abs/2304.03153.
>
> Shijie Geng, Shuchang Liu, Zuohui Fu, Yingqiang Ge, and Yongfeng Zhang. Recommendation as Language Processing (RLP): A Unified Pretrain, Personalized Prompt & Predict Paradigm (P5). In Proceedings of the 16th ACM Conference on Recommender Systems, pp. 299–315, Seattle WA USA, September 2022. ACM. ISBN 978-1-4503-9278-5. doi: 10.1145/3523227.3546767.
>
> Jinming Li, Wentao Zhang, Tian Wang, Guanglei Xiong, Alan Lu, and Gérard G. Medioni. GPT4Rec: A Generative Framework for Personalized Recommendation and User Interests Interpretation. In Surya Kallumadi, Yubin Kim, Tracy Holloway King, Shervin Malmasi, Maarten de Rijke, and Jacopo Tagliabue (eds.), Proceedings of the 2023 SIGIR Workshop on eCommerce co-located with the 46th International ACM SIGIR Conference on Research and Development in Information Retrieval (SIGIR 2023), Taipei, Taiwan, July 27, 2023, volume 3589 of CEUR Workshop Proceedings. CEUR-WS.org, 2023.
>
> Lei Li, Yongfeng Zhang, Dugang Liu, and Li Chen. Large language models for generative recommendation: A survey and visionary discussions. In Nicoletta Calzolari, Min-Yen Kan, Veronique Hoste, Alessandro Lenci, Sakriani Sakti, and Nianwen Xue (eds.), Proceedings of the 2024 Joint International Conference on Computational Linguistics, Language Resources and Evaluation (LREC-COLING 2024), pp. 10146–10159, Torino, Italia, May 2024. ELRA and ICCL.

---

### Review · Reviewer_N4uZ · 2026-03-31

**Summary Of Contributions:**

This paper presents a reproduction study of FACTER, a recent ICML’25 framework for fairness-aware LLM-based recommendation using conformal thresholding and iterative prompt repair. The authors identify discrepancies between reported and reproduced performance, and analyze the underlying causes. In addition, they propose a re-ranking formulation of the problem, which enables more controlled evaluation and practical deployment.

Strengths

- Careful diagnosis of issues in the original FACTER framework
    - *Performance gains:* The paper provides a detailed breakdown showing that the dynamic repair loop does not significantly reduce model bias, but instead relaxes the evaluation threshold over time, leading to fewer reported violations.
    - *Performance gap:* The reproduced results under open-ended generation differ substantially from those reported in the original paper, suggesting sensitivity to evaluation protocol details and instability in unconstrained generation.
- Improved experimental clarity and reproducibility
    - The proposed re-ranking formulation simplifies the evaluation setting, making the problem more controlled and closer to practical recommendation pipelines.
    - The authors provide an open-source reimplementation, which improves transparency and reproducibility for future work.

Weakness

- The evaluation depends on semantic similarity from MPNet embeddings, which may itself introduce model bias and may not fully capture meaningful notions of fairness.
- The re-ranking setup assumes the availability of a relevant candidate set, focusing only on ranking. However, fairness issues may also arise in the retrieval stage, which is not addressed.

**Audience:**

Yes

**Audience Explanation:**

This paper should be of interest to researchers working on fairness in recommender systems, LLM-based recommendation. The study highlights how performance gains can depend on underspecified implementation details and evaluation protocols, which is an important issue for the broader community. By providing a clearer and more reproducible framework, the paper contributes to improving experimental rigor and transparency in this area.

**Claims And Evidence:**

Yes

**Claims Explanation:**

The paper provides extensive empirical evidence and implementation details to support its claims regarding reproducibility issues in FACTER. The comparison between open-ended generation and re-ranking settings is well-motivated, and the proposed re-ranking framework enables more controlled analysis. In particular, the breakdown of different components (adaptive thresholding vs. prompt repair) is clearly presented and supports the claim that performance gains are largely driven by threshold dynamics.

There remain some minor concerns regarding fairness evaluation and the assumptions introduced by the re-ranking framework, but these do not substantially weaken the overall empirical findings. The conclusions are generally convincing, though some interpretations could be framed more cautiously.

**Requested Changes:**

Building on the weaknesses mentioned above, I suggest the following improvements:

- Provide a more detailed discussion of the limitations of embedding-based fairness metrics, including potential bias introduced by the encoder.
- Discuss the role of retrieval-stage fairness, which is not covered in the current re-ranking setup.

---

> ### Author Response · Authors · 2026-04-10
> **Addressing bias in the reranking pipeline and embedding-based metrics**
>
> Thank you for your constructive feedback and for acknowledging the value of our "careful diagnosis" of the FACTER framework. We appreciate the opportunity to strengthen the discussion regarding the limitations of the current evaluation pipeline.
> In response to your suggestions, we have implemented the following adjustments:
>
> **1: Discussion on Embedding-Based Metric Limitations.** We have added a discussion in Section 6.2 regarding the potential biases inherent in the MPNet encoder. We now explicitly state that since all fairness metrics are computed in the representation space of this encoder, any learned biases resulting from demographic-content interactions in its training data will propagate into the calibration thresholds and evaluation scores. We acknowledge that while the original study addresses random perturbations, it does not account for systematic encoder bias, a limitation our reproduction inherits.
>
> **2: Role of Retrieval-Stage Fairness.** We have expanded our discussion in Section 6.2 to address the scope of the re-ranking formulation. We clarify that our setup assumes the availability of a candidate set containing relevant items, which in practice is produced by an upstream retriever. We now explicitly discuss that demographic bias occurring at the retrieval stage cannot be corrected by downstream re-ranking alone. We conclude that end-to-end fairness requires coordinated interventions across every stage of the recommendation pipeline, as studied by Hsu et al., 2025.
>
> **3: Reframing and Factual Reporting.** Consistent with feedback across the review panel, we have reframed our main claims to be more factual. We emphasize that our best-effort re-implementation shows that results are highly sensitive to implementation and evaluation details. We focus our results on the finding that FACTER's apparent gains are largely driven by adaptive threshold relaxation, rather than absolute bias erasure.
>
> **4: Documentation of Missing Artifacts.** To address concerns regarding the reproducibility landscape, we have consolidated all identified ambiguities and our corresponding implementation choices into a comprehensive audit table in Appendix B. This includes the multi-target setup, threshold dynamics, and metric definitions, providing a clear "Single Source of Truth" for the discrepancies encountered during our study.
>
> References:
> Brian Hsu, Cyrus DiCiccio, Natesh S. Pillai, and Hongseok Namkoong. From models to systems: A comprehensive framework for ai system fairness in compositional recommender systems. In Miriam Rateike, Awa Dieng, Jamelle Watson-Daniels, Ferdinando Fioretto, and Golnoosh Farnadi (eds.), Proceedings of the Algorithmic Fairness Through the Lens of Metrics and Evaluation, volume 279 of Proceedings of Machine Learning Research, pp. 8–37. PMLR, 14 Dec 2025. URL https://proceedings.mlr.press/v279/hsu25a.html.

---

### Author Response · Authors · 2026-06-07
**Thank you**

We want to thank the reviewers and the Action Editor for the careful assessment of our manuscript and for the feedback throughout the review process. We appreciate the time and effort spent evaluating the work, and we are grateful for the final recommendation.

---

### Decision · Action_Editor_V5E4 · 2026-05-16

**Recommendation:** Accept as is

**Audience:**

Yes

**Audience Explanation:**

Reviewers acknowledged that this is a useful reproduction study showing that FACTER’s open-generation utility results are hard to reproduce, and that its fairness gains seem largely driven by adaptive threshold relaxation rather than clear bias removal.

**Claims And Evidence:**

Yes

**Claims Explanation:**

This paper presents a careful reproduction study of FACTER for fair LLM-based recommendation and adds a re-ranking extension to better understand the method in a constrained setting.

Certain reviewers raised limitations, especially around the re-ranking setup, MPNet-based metrics, and the unreproduced UP5 baseline. But they believed the revised version now states these issues more clearly and keeps the claims appropriately qualified. Therefore I think there is NO outstanding issues.